# Deep Active Learning by Leveraging Training Dynamics

## Abstract

Active learning theories and methods have been extensively studied in classical statistical learning settings. However, deep active learning, i.e., active learning with deep learning models, is usually based on empirical criteria without solid theoretical justification, thus suffering from heavy doubts when some of those fail to provide benefits in applications. In this paper, by exploring the connection between the generalization performance and the training dynamics, we propose a theory-driven deep active learning method (***dynamicAL***) which selects samples to maximize training dynamics. In particular, we prove that convergence speed of training and the generalization performance is positively correlated under the ultra-wide condition and show that maximizing the training dynamics leads to a better generalization performance. Further on, to scale up to large deep neural networks and data sets, we introduce two relaxations for the subset selection problem and reduce the time complexity from polynomial to constant. Empirical results show that *dynamicAL* not only outperforms the other baselines consistently but also scales well on large deep learning models. We hope our work inspires more attempts in bridging the theoretical findings of deep networks and practical impacts in deep active learning applications.

## 1 Introduction

Training deep learning (DL) models usually requires a large amount of high-quality labeled data (Zhang et al., 2017) to optimize a model with a massive number of parameters. The acquisition of such annotated data is usually time-consuming and expensive, making it unaffordable in the fields that require high domain expertise. A promising approach for minimizing the labeling effort is active learning (AL), which aims to identify and label the maximally informative samples, so that a high-performing classifier can be trained with minimal labeling effort (Settles, 2009). Under classical statistical learning settings, theories of active learning have been extensively studied from the perspective of VC dimension (Hanneke et al., 2014). As a result, a variety of methods have been proposed, such as (i) the version-space-based approach, which requires maintaining a set of models (Cohn et al., 1994; Balcan et al., 2009), and (ii) the clustering-based approach, which assumes the data within the same cluster have pure labels (Dasgupta & Hsu, 2008).

However, the theoretical analyses for these classical settings may not hold for over-parameterized deep neural networks where the traditional wisdom is ineffective (Zhang et al., 2017). For example, margin-based methods select the labeling examples in the vicinity of the learned decision boundary (Balcan et al., 2007; Balcan & Long, 2013). However, in the over-parameterized regime, every labeled example could potentially be near the learned decision boundary (Karzand & Nowak, 2019). As a result, theoretically, such analysis can hardly guide us to design practical active learning methods. Besides, empirically, multiple deep active learning works, borrowing observations and insights from the classical theories and methods, have been observed unable to outperform their passive learning counterparts in a few application scenarios (Kirsch et al., 2019; Ash et al., 2020).

On the other hand, the analysis of neural network's optimization and generalization performance has witnessed several exciting developments in recent years in terms of the deep learning theory (Jacot et al., 2018; Arora et al., 2019b; Lee et al., 2019). It is shown that the training dynamics of deep neural networks using gradient descent can be characterized by the Neural Tangent Kernel (NTK) of infinite (Jacot et al., 2018) or finite (Hanin & Nica, 2020) width networks. This is further leveraged

to characterize the generalization of over-parameterized networks through Rademacher complexity analysis (Arora et al., 2019b; Cao & Gu, 2019). We are therefore inspired to ask: How can we design a practical and generic active learning method for deep neural networks with theoretical justifications?

To answer this question, we firstly explore the connection between the model performance on testing data and the convergence speed on training data for the over-parameterized deep neural networks. Based on the NTK framework (Jacot et al., 2018; Arora et al., 2019b), we theoretically show that if a deep neural network converges faster ("Train Faster"), then it tends to have a better generalization performance ("Generalize Better"), which matches with the existing observations (Hardt et al., 2016; Liu et al., 2017; Lyle et al., 2020; Ru et al., 2020; Xu et al., 2021). Motivated by the aforementioned connection, we first introduce *Training Dynamics*, the derivative of training loss with respect to iteration, as a proxy to quantitatively describe the training process. On top of it, we formally propose our generic and theoretically-motivated deep active learning method, *dynamicAL*, which will query labels for a subset of unlabeled samples that maximally increases the training dynamics. In order to compute the training dynamics by merely using the unlabeled samples, we leverage two relaxations *Pseudo-labeling* and *Subset Approximation* to solve this non-trivial subset selection problem. Our relaxed approaches are capable of effectively estimating the training dynamics as well as efficiently solving the subset selection problem by reducing the complexity from $O(N^b)$ to $O(b)$.

In theory, we coin a new term *Alignment* to measure the length of the label vector's projection on the neural tangent kernel space. Then, we demonstrate that higher alignment usually comes with a faster convergence speed and a lower generalization bound. Furthermore, we extend the previous analysis to an active learning setting, where the i.i.d. assumption may not hold with the help of the maximum mean discrepancy (Borgwardt et al., 2006). Finally, we show that alignment is positively correlated with our active learning goal, training dynamics, which implies that maximizing training dynamics will lead to a better generalization performance.

Regarding experiments, we have empirically verified our theory by conducting extensive experiments on three datasets, CIFAR10 (Krizhevsky et al., 2009), SVHN (Netzer et al., 2011), and Caltech101 (Fei-Fei et al., 2004) using three types of network structure: vanilla CNN, ResNet (He et al., 2016), and VGG (Simonyan & Zisserman, 2015). We first show that result of the subset selection problem delivered by the subset approximation is close to the global optimum solution. Furthermore, under active learning setting, our method not only outperforms the other baselines but also scales well on large deep learning models.

The main contributions of our paper can be summarized as follows:

- We propose a theory-driven deep active learning method, *dynamicAL*, inspired by the results of "train faster, generalize better". To this end, we introduce the Training Dynamics, as a proxy to describe the training process and embody the inspiration.

- We demonstrate that the convergence speed of training and the generalization performance is (positively) strongly correlated under the ultra-wide condition; we also show that maximizing the training dynamics will lead to a lower generalization error in the scenario of active learning.

- Our method is easy to implement. We conduct extensive experiments to evaluate the effectiveness of *dynamicAL* and empirically show that our method consistently outperforms other methods in a wide range of active learning settings.

## 2 BACKGROUND

### 2.1 NOTATIONS

We use the random variable $x \in \mathcal{X}$ to represent input data feature and $y \in \mathcal{Y}$ as label where $K$ is the number of classes and $[K] := \{1, 2, ..., K\}$. We are given a data source $D$ with unknown distribution $p(x, y)$. We further denote the concatenation of $x$ as $X = [x_1, x_2, ..., x_M]^\top$ and that of $y$ as $Y = [y_1, y_2, ..., y_M]^\top$. We consider a deep learning classifier $h_\theta(x) = \arg\max \sigma(f(x; \theta)) : x \to y$ parameterized by $\theta \in \mathbb{R}^p$, where $\sigma(\cdot)$ is the softmax function and $f$ is a neural network. Let $\otimes$ be the Kronecker Product and $I_K \in \mathbb{R}^{K \times K}$ be an identity matrix.

### 2.2 ACTIVE LEARNING

The goal of active learning is to improve the learning efficiency of a model with a limited labeling budget. In this work, we consider the pool-based AL setup, where a finite data set $S = \{(x_l, y_l)\}_{l=1}^M$

with $M$ points are $i.i.d.$ sampled from the $p(x, y)$ as the (initial) labeled set. The AL model receives an unlabeled data set $U$ sampled from $p(x)$ and request labels according to $p(y|x)$ for any $x \in U$ in each query round. There are $R$ rounds in total and for each round, a query set $Q$ consisting of $b$ unlabeled samples can be queried. The total budget size $B = b \times R$.

## 2.3 NEURAL TANGENT KERNEL

The Neural Tangent Kernel (Jacot et al., 2018) has been widely applied to analyze the dynamics of neural networks. If the neural network is sufficiently wide, properly initialized, and trained by gradient descent with infinitesimal step size (a.k.a. gradient flow), then the neural network is equivalent to kernel regression predictor with a deterministic kernel $\Theta(\cdot, \cdot)$, called Neural Tangent Kernel (NTK). When minimizing the mean squared error loss, at the iteration $t$, the dynamics of the neural network $f$ has a closed-form expression:

$$\frac{df(\mathcal{X}; \theta(t))}{dt} = -\mathcal{K}_t(\mathcal{X}, \mathcal{X})\left(f(\mathcal{X}; \theta(t)) - \mathcal{Y}\right) \tag{1}$$

where $\theta(t)$ denotes the parameter of the neural network at iteration $t$, $\mathcal{K}_t(\mathcal{X}, \mathcal{X}) \in \mathbb{R}^{|\mathcal{X}| \times K \times |\mathcal{X}| \times K}$ is called the empirical NTK and $\mathcal{K}_t^{i,j}(x, x') = \nabla_\theta f^i(x; \theta(t))^\top \nabla_\theta f^j(x'; \theta(t))$, for two samples $x, x' \in \mathcal{X}$ and $i, j \in [K]$. The time-variant kernel $\mathcal{K}_t(\cdot, \cdot)$ is equivalent to the (time-invariant) NTK with high probability, that is, if the neural network is sufficiently wide and properly initialized, then:

$$\mathcal{K}_t(\mathcal{X}, \mathcal{X}) = \Theta(\mathcal{X}, \mathcal{X}) \otimes I_K \tag{2}$$

The final learned neural network at iteration $t$, is equivalent to the kernel regression solution with respect to the NTK (Lee et al., 2019). For any input $x$ and training data $\{X, Y\}$ we have,

$$f(x; \theta(t)) \approx \Theta(x, X)^\top \Theta(X, X)^{-1}(I - e^{-\eta\Theta(X, X)t})Y \tag{3}$$

where $\eta$ is the learning rate, $\Theta(x, X)$ is the NTK matrix between input $x$ and all samples in training data $X$.

## 3 METHOD

In section 3.1, we introduce the notion of training dynamics which can be used to describe the training process. Then, in section 3.2, based on the training dynamics, we proposed *dynamicAL*. In section 3.3, we discuss the connection between *dynamicAL* and existing deep active learning methods.

## 3.1 TRAINING DYNAMICS

In this section, we introduce the notion of training dynamics. The cross-entropy loss over labeled set $S$ is defined as:

$$L(S) = \sum_{(x_l, y_l) \in S} \ell(f(x_l; \theta), y_l) = -\sum_{(x_l, y_l) \in S} \sum_{i \in [K]} y_l^i \log \sigma^i(f(x_l; \theta)) \tag{4}$$

where $\sigma^i(f(x; \theta)) = \frac{\exp(f^i(x; \theta))}{\sum_j \exp(f^j(x; \theta))}$. We first analyze the dynamics of the training loss, with respect to iteration $t$, on one labeled sample (derivation is in Appendix A.1):

$$\frac{\partial \ell(f(x; \theta), y)}{\partial t} = -\sum_i \left(y^i - \sigma^i(f(x; \theta))\right)\nabla_\theta f^i(x; \theta)\nabla_t^\top \theta \tag{5}$$

For neural networks trained by gradient descent, if learning rate $\eta$ is small, then $\nabla_t \theta = \theta_{t+1} - \theta_t = -\eta\frac{\partial \sum_{(x_l, y_l) \in S} \ell(f(x_l; \theta), y_l)}{\partial \theta}$. Taking the partial derivative of the training loss with respect to the parameters, we have (the derivation of the following equation can be found in Appendix A.2):

$$\frac{\partial \ell(f(x; \theta), y)}{\partial \theta} = \sum_{j \in [K]} \left(\sigma^j(f(x; \theta)) - y^j\right)\frac{\partial f^j(x; \theta)}{\partial \theta} \tag{6}$$

Therefore, we can further get the following result for the dynamics of training loss,

$$\frac{\partial \ell(f(x; \theta), y)}{\partial t} = -\eta \sum_i \left(\sigma^i(f(x; \theta)) - y^i\right) \sum_j \sum_{(x_{l'}, y_{l'}) \in S} \nabla_\theta f^i(x; \theta)^\top \nabla_\theta f^j(x_{l'}; \theta)\left(\sigma^j(f(x_{l'}; \theta)) - y_{l'}^j\right) \tag{7}$$

Furthermore, we define $d^i(X, Y) = \sigma^i(f(X; \theta)) - Y^i$ and $Y^i$ is the label vector of all samples for $i$-th class. Then, the *training dynamics* (dynamics of training loss) over training set $S$ is denoted by $G(S) \in \mathbb{R}$:

$$G(S) = -\frac{1}{\eta} \sum_{(x_l, y_l) \in S} \frac{\partial \ell(f(x_l; \theta), y_l)}{\partial t} = \sum_i \sum_j d^i(X, Y)^\top \mathcal{K}^{ij}(X, X) d^j(X, Y) \qquad (8)$$

## 3.2 ACTIVE LEARNING BY ACTIVATING TRAINING DYNAMICS

Before we present *dynamicAL*, we state the Proposition 1, which serves as theoretical guidance for *dynamicAL* and will be proved in Section 4.

**Proposition 1.** *For deep neural networks, converge faster leads to a lower worst-case generalization error.*

Motivated by the connection between convergence speed and generalization performance, we propose the general-purpose active learning method, *dynamicAL*, which aims to accelerate the convergence by querying labels for unlabeled samples. As we described in the previous section, the training dynamics can be used to describe the training process. Therefore, we employ the training dynamics as a proxy to design an active learning method. Specifically, at each query round, *dynamicAL* will query labels for samples which maximize the training dynamics $G(S)$, *i.e.*,

$$Q = \text{argmax}_{Q \subseteq U} G(S \cup \overline{Q}), \ s.t. \ |Q| = b \qquad (9)$$

where $\overline{Q}$ is the corresponding data set for $Q$ with ground-truth labels. Notice that when applying the above objective in practice, we are facing two major challenges. First, $G(S \cup \overline{Q})$ cannot be directly computed, because the label information of unlabeled examples is not available before the query. Second, the subset selection problem can be computationally prohibitive if enumerating all possible sets with size $b$. Therefore, we employ the following two relaxations to make this maximization problem to be solved with constant time complexity.

**Pseudo Labeling.** To estimate the training dynamics, we use the predicted label $\hat{y}_u$ for sample $x_u$ in the unlabeled data set $U$ to compute $G$. Note, the effectiveness of this adaptation has been demonstrated in the recent gradient-based methods (Ash et al., 2020; Mu et al., 2020), which compute the gradient as if the model's current prediction on the example is the true label. Therefore, the maximization problem in Equation (9) is changed to,

$$Q = \text{argmax}_{Q \subseteq U} G(S \cup \widehat{Q}) \qquad (10)$$

where $\widehat{Q}$ is the corresponding data set for $Q$ with peseudo labels, $\widehat{Y}_Q$.

**Subset Approximation.** The subset selection problem of Equation (10) still requires enumerating all possible subsets of $U$ with size $b$, which is $O(n^b)$. We simplify the selection problem to the following problem without causing any change on the result,

$$\text{argmax}_{Q \subseteq U} G(S \cup \widehat{Q}) = \text{argmax}_{Q \subseteq U} \Delta(\widehat{Q}|S) \qquad (11)$$

where $\Delta(\widehat{Q}|S) = G(S \cup \widehat{Q}) - G(S)$ is defined as the change of training dynamics. We approximate the change of training dynamics caused by query set $Q$ using the summation of the change of training dynamics caused by each sample in the query set. Then the maximization problem can be converted to Equation (12) which can be solved by a greedy algorithm with $O(b)$.

$$Q = \text{argmax}_{Q \subseteq U} \sum_{(x_u, \widehat{y}_u) \in \widehat{Q}} \Delta(\{(x_u, \widehat{y}_u)\}|S), \ s.t. \ |Q| = b \qquad (12)$$

To further show the approximated result is reasonably good, we decompose the change of training dynamics as (derivation in Appendix A.4) ,

$$\Delta(\widehat{Q}|S) = \sum_{(x_u, \widehat{y}_u) \in \widehat{Q}} \Delta(\{(x_u, \widehat{y}_u)\}|S) + \sum_{(x_u, \hat{y}_u), (x_{u'}, \hat{y}_{u'}) \in \widehat{Q}} d^i(x_u, \hat{y}_u)^\top \mathcal{K}^{ij}(x_u, x_{u'}) d^i(x_{u'}, \hat{y}_{u'}) \qquad (13)$$

The first term in the right hand side is the approximated change of training dynamics. Then, we further define the *Approximation Ratio* (14) which measures the approximation quality,

$$R(\widehat{Q}|S) = \frac{\sum_{(x_u, \widehat{y}_u) \in \widehat{Q}} \Delta(\{(x_u, \widehat{y}_u)\}|S)}{\Delta(\widehat{Q}|S)} \qquad (14)$$

We empirically measure the expectation of the Approximation Ratio on three data sets with three different neural networks. As shown in Figure 4, the expectation, $\mathbb{E}_{Q \sim U} R(\widehat{Q}|S) \approx 1$ when the model is converged. Therefore, the approximated result delivered by greedy algorithm is close to the global optimal solution of the original maximization problem, Equation (10), especially when the model is converged.

Based on the above two approximations, we present the proposed method *dynamicAL* in Algorithm 1. As described below, the algorithm starts by training a neural network $f(\cdot; \theta)$ on initial labeled set $S$ until converge. Then, for every unlabeled sample $x_u$, we compute pseudo label $\hat{y}_u$ and the change of training dynamics $\Delta(\{(x_u, \widehat{y}_u)\}|S)$. After that, *dynamicAL* will query labels for top-$b$ samples causing maximal change on training dynamics. Then train the neural network (without re-initialization) on the extended labeled set, and repeat the process.

---

**Algorithm 1** Deep Active Learning by Leveraging Training Dynamics

**Input:** Neural network $f(\cdot; \theta)$, unlabeled sample set $U$, initial labeled set $S$, number of query round $R$, query batch size $b$.
**for** $r = 1$ **to** $R$ **do**
    Train $f(\cdot; \theta)$ on $S$ with cross-entropy loss until convergence.
    **for** $x_u \in U$ **do**
        Compute its pseudo label $\hat{y}_u = \text{argmax} f(x_u; \theta)$.
        Compute $\Delta\left(\{(x_u, \widehat{y}_u)\}|S\right)$.
    **end for**
    Select $b$ query samples $Q$ with the highest $\Delta$ values, and request their labels from the oracle.
    Update the labeled data set $S = S \cup Q$ .
**end for**
**return** Final model $f(\cdot; \theta)$.

---

## 3.3 RELATION TO EXISTING METHODS

Although existing deep active learning methods are usually designed based on heuristic criteria, some of them have empirically shown their effectiveness (Ash et al., 2020; Liu et al., 2021; Huang et al., 2016). We surprisingly found that our theoretically-motivated method *dynamicAL* has some connections with those existing methods from the perspective of active learning criterion. The proposed active learning criterion in Equation (12) can be explicitly written as (derivation in Appendix A.5) :

$$\Delta(\{(x_u, \widehat{y}_u)\}|S) = \|\nabla_\theta \ell(f(x_u; \theta), \hat{y}_u)\|^2 + 2 \sum_{(x,y) \in S} \nabla_\theta \ell(f(x_u; \theta), \hat{y}_u)^\top \nabla_\theta \ell(f(x; \theta), y) \tag{15}$$

The first term of right-hand side can be interpreted as the square of gradient length (2-norm) which reflects the uncertainty of the model on the example and has been wildly used as an active learning criterion in some existing works (Huang et al., 2016; Ash et al., 2020; Shukla, 2021). The second term can be regarded as the influence function (Koh & Liang, 2017) with identity hessian matrix. And recently, Liu et al. (2021) has empirically shown that the effectiveness of using it as the active learning criterion. We hope our theoretical analysis can also shed some light on the interpretation of previous methods.

## 4 THEORETICAL ANALYSIS

In this section, we study the correlation between the convergence rate of the training loss and the generalization error under the ultra-wide condition (Jacot et al., 2018; Arora et al., 2019b). We define a measure named *alignment* to quantify the convergence rate and further show the connection with generalization bound. The analysis provides a theoretical guarantee for the phenomenon of "Train Faster, Generalization Better" as well as our active learning method *dynamicAL* with a rigorous treatment. Finally, we show that the our active learning proxy, training dynamics, is correlated with alignment, which indicates that increasing the training dynamics leads to larger convergence rate and better generalization performance. We leave all proofs of theorems and details of verification experiments in Appendix B and D respectively.

### 4.1 TRAIN FASTER PROVABLY GENERALIZE BETTER

Given an ultra-wide neural network, the gradient descent can achieve a near-zero training error (Jacot et al., 2018; Arora et al., 2019a) and its generalization ability in unseen data can be bounded (Arora et al., 2019b). It is shown that both convergence and generalization have a connection with the NTK (Arora et al., 2019b). However, the question what is the relation between convergence rate and generalization bound has not been answered. We formally give a solution by introducing the concept of *alignment*, which is defined as follows,

**Definition 1** (Alignment). *Given a data set $S = \{X, Y\}$, the alignment is a measure of correlation between $X$ and $Y$ projected in the NTK space. In particular, the alignment can be computed by $\mathcal{A}(X, Y) = \text{Tr}[Y^\top \Theta(X, X)Y] = \sum_{k=1}^{K} \sum_{i=1}^{n} \lambda_i (\vec{v}_i^\top Y^k)^2$.*

In the following, we will demonstrate why train faster generalizes better through alignment. In particular, the relation of convergence rate and generalization bound with alignment are analyzed. The convergence rate of gradient descent for ultra-wide networks is presented in following lemma,

**Lemma 1** (Convergence Analysis with NTK, Theorem 4.1 of (Arora et al., 2019b)). *Suppose $\lambda_0 = \lambda_{\min}(\Theta) > 0$ for all subsets of data samples. For $\delta \in (0, 1)$, if $m = \Omega(\frac{n^7}{\lambda_0^4 \delta^4 \epsilon^2})$ and $\eta = O(\frac{\lambda_0}{n^2})$. Then with probability at least $1 - \delta$, the network can achieve near-zero training error,*

$$\|Y - f(X; \theta(t))\|_2 = \sqrt{\sum_{k=1}^{K} \sum_{i=1}^{n} (1 - \eta \lambda_i)^{2t} (\vec{v}_i^\top Y^k)^2} \pm \epsilon \tag{16}$$

*where $n$ denotes the number of training samples, $m$ the width of hidden layers. The NTK $\Theta = V^\top \Lambda V$ with $\Lambda = \{\lambda_i\}_{i=1}^{n}$ a diagonal matrix of eigenvalues and $V = \{\vec{v}_i\}_{i=1}^{n}$ a unitary matrix.*

In this lemma, we take mean square error (MSE) loss as an example for the convenience of illustration. The conclusion can be extended to other loss functions such as cross-entropy loss (see Appendix B.2 in (Lee et al., 2019)). From the lemma, we find convergence rate is governed by the dominant term (16) as $\mathcal{E}_t(X, Y) = \sqrt{\sum_{k=1}^{K} \sum_{i=1}^{n} (1 - \eta \lambda_i)^{2t} (\vec{v}_i^\top Y^k)^2}$, which is correlated with the *alignment*:

**Theorem 1** (Relationship between convergence rate and alignment). *Under the same assumptions as in Lemma 1. The convergence rate described by $\mathcal{E}_t$ satisfies,*

$$\text{Tr}[Y^\top Y] - 2t\eta \mathcal{A}(X, Y) \le \mathcal{E}_t^2(X, Y) \le \text{Tr}[Y^\top Y] - \eta \mathcal{A}(X, Y) \tag{17}$$

**Remark 1.** *In the above theorem, we demonstrate that the alignment can measure the convergence rate. Especially, we find that both the upper bound and the lower bound of error $\mathcal{E}_t(X, Y)$ are inversely proportional to the alignment, which implies that higher alignment will lead to achieving faster convergence.*

Now we analyze the generalization performance of the proposed method through complexity analysis. We demonstrate that the ultra-wide networks can achieve a reasonable generalization bound.

**Lemma 2** (Generalization bound with NTK, Theorem 5.1 of (Arora et al., 2019b)). *Suppose data $S = \{(x_i, y_i)\}_{i=1}^{n}$ are i.i.d. samples from a non-degenerate distribution $p(x, y)$, and $m \ge \text{poly}(n, \lambda_0^{-1}, \delta^{-1})$. Consider any loss function $\ell : \mathbb{R} \times \mathbb{R} \to [0, 1]$ that is 1-Lipschitz. Then with probability at least $1 - \delta$ over the random initialization, the network trained by gradient descent for $T \ge \Omega(\frac{1}{\eta \lambda_0} \log \frac{n}{\delta})$ iterations has population risk $\mathcal{L}_p = \mathbb{E}_{(x,y) \sim p(x,y)}[\ell(f_T(x; \theta), y)]$ that is bounded as follows:*

$$\mathcal{L}_p \le \sqrt{\frac{2\text{Tr}[Y^\top \Theta^{-1}(X, X)Y]}{n}} + O\left(\sqrt{\frac{\log \frac{n}{\lambda_0 \delta}}{n}}\right). \tag{18}$$

In this lemma, we show that the dominant term in the generalization upper bound is $\mathcal{B}(X, Y) = \sqrt{\frac{2\text{Tr}[Y^\top \Theta^{-1} Y]}{n}}$. In the following theorem, we further prove that this bound is inversely proportional to the alignment $\mathcal{A}(X, Y)$.

**Theorem 2** (Relationship between the generalization bound and alignment). *Under the same assumptions as in Lemma 2. If we define the generalization upper bound as $\mathcal{B}(X, Y) = \sqrt{\frac{2\text{Tr}[Y^\top \Theta^{-1} Y]}{n}}$, then it can be bounded with the alignment as follows,*

$$\frac{\text{Tr}^2[Y^\top Y]}{\mathcal{A}(X, Y)} \le \frac{n}{2} \mathcal{B}^2(X, Y) \le \frac{\lambda_{max}}{\lambda_{min}} \frac{\text{Tr}^2[Y^\top Y]}{\mathcal{A}(X, Y)} \tag{19}$$

**Remark 2.** *Theorems 1 and 2 reveal that the cause for the correlated phenomenons "Train Faster" and "Generalize Better" is the projection of label vector on the NTK space (alignment).*

## 4.2 "TRAIN FASTER, GENERALIZE BETTER" FOR ACTIVE LEARNING

In the NTK framework (Arora et al., 2019b), the empirical average requires data in $S$ is *i.i.d.* samples, Lemma 2. However, this assumption may not hold in the active learning setting with multiple query rounds, because the training data is composed by *i.i.d.* sampled initial label set and samples queried by active learning policy. To extend the previous analysis principle to active learning, we follow (Wang & Ye, 2015) to reformulate the Lemma 2 as:

$$\mathcal{L}_p \leq (\mathcal{L}_p - \mathcal{L}_q) + \sqrt{\frac{2\,\mathrm{Tr}[Y^\top \mathbf{\Theta}^{-1}(X,X)Y]}{n}} + O\left(\sqrt{\frac{\log \frac{n}{\lambda_0 \delta}}{n}}\right). \tag{20}$$

where $\mathcal{L}_q = \mathbb{E}_{(x,y)\sim q(x,y)}[\ell(f(x;\theta),y)]$, $q(x,y)$ denotes the data distribution after query, and $X, Y$ includes initial training samples and samples after query. There is a new term in the upper bound, which is the difference between the true risk under different data distributions.

$$\mathcal{L}_p - \mathcal{L}_q = \mathbb{E}_{(x,y)\sim p(x,y)}[\ell(f(x;\theta),y)] - \mathbb{E}_{(x,y)\sim q(x,y)}[\ell(f(x;\theta),y)] \tag{21}$$

Though in active learning the data distribution for the labeled samples may be different from the original distribution. However, they share the same conditional probability $p(y|x)$. We define $g(x) = \int_y \ell(f(x;\theta),y)p(y|x)dy$. Then we have,

$$\mathcal{L}_p - \mathcal{L}_q = \int_x g(x)p(x)dx - \int_x g(x)q(x)dx \tag{22}$$

To measure the distance between two distribution, we employ the Maximum Mean Discrepancy (MMD) with neural tangent kernel (Jia et al., 2021) (derivation in Appendix B.3).

$$\mathcal{L}_p - \mathcal{L}_q \leq \mathrm{MMD}(S_0, S, \mathcal{H}_\mathbf{\Theta}) + O\left(\sqrt{\frac{C\ln(1/\delta)}{n}}\right) \tag{23}$$

Slightly overloading the notation, we denote the initial labeled set as $S_0$, $\mathcal{H}_\mathbf{\Theta}$ as the associated reproducing kernel Hilbert space for the NTK $\mathbf{\Theta}$, and $\forall x, x' \in S, \mathbf{\Theta}(x,x') \leq C$. Note, $\mathrm{MMD}(S_0, S, \mathcal{H}_\mathbf{\Theta})$ as the empirical measure for $\mathrm{MMD}(p(x), q(x), \mathcal{H}_\mathbf{\Theta})$. We empirically compute MMD and the dominant term of the generalization upper bound $\mathcal{B}$ under the active learning setting with our method *dynamicAL*. As shown in Figure 1, on CIFAR10 with a CNN target model (three convolutional layers with global average pooling), the initial labeled set size $|S| = 500$, query round $R = 1$ and budget size

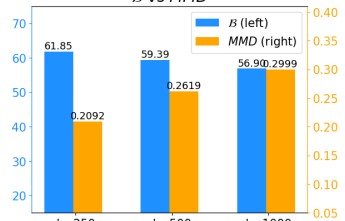

Figure 1: Empirical Generalization Bound and MMD

$b \in \{250, 500, 1000\}$, we observe that, under different active learning setting, the MMD is always much smaller than the $\mathcal{B}$. Therefore, the lemma 2 still holds for the target model with *dynamicAL*. The result for $R \geq 2$ is in Appendix E.4. The computation detail of MMD and NTK is in Appendix D.1.

## 4.3 ALIGNMENT AND TRAINING DYNAMICS IN ACTIVE LEARNING SETTING

In this section, we show the relationship between the alignment and the training dynamics. To be consistent with the previous theoretical analysis (Theorem 1 and 2), we use the training dynamics with mean square error under the ultra-width condition, which can be expressed as $G_{MSE}(S) = \mathrm{Tr}\left[(f(X;\theta) - Y)^\top \mathbf{\Theta}(X,X)(f(X;\theta) - Y)\right]$. Due to the limited space, we leave the derivation in Appendix A.3. To further quantitatively evaluate the correlation between $G_{MSE}(S \cup \overline{Q})$ and $\mathcal{A}(X\|X_Q, Y\|Y_Q)$, we utilize the Kendall $\tau$ coefficient (Kendall, 1938) to empirically measure their relation. As shown in Figure 2, for CNN on CIFAR10 with active learning setting, where $|S| = 500$ and $|\overline{Q}| = 250$, there is a strong agreement between $G_{MSE}(S \cup \overline{Q})$

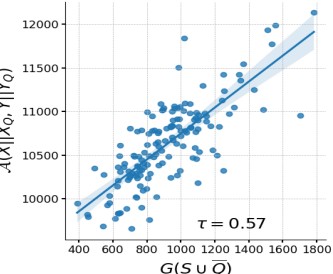

Figure 2: Alignment and Training Dynamics.

and $\mathcal{A}(X\|X_Q, Y\|Y_Q)$, which further indicates that increasing the training dynamics will lead to a faster convergence and better generalization performance. More details about this verification experiment are in Appendix D.2.

# 5 EXPERIMENTS

## 5.1 EXPERIMENT SETUP

**Baselines.** We compare *dynamicAL* with the following eight baselines: Random, Corset, Confidence Sampling (Conf), Margin Sampling (Marg), Entropy, and Active Learning by Learning (ALBL), Batch Active learning by Diverse Gradient Embeddings (BADGE). Description of baseline methods is in Appendix E.1.

**Data set and Target Model.** We evaluate all the methods on three benchmark data sets, namely, CIFAR10 (Krizhevsky et al., 2009), SVHN (Netzer et al., 2011), and Caltech101 (Fei-Fei et al., 2004). We use the accuracy as the evaluation metric and report the mean value of 5 runs. We consider three neural network architectures: vanilla CNN, ResNet18 (He et al., 2016), and VGG11 (Simonyan & Zisserman, 2015). For each model, we keep the hyper-parameters used in their official implementations. More information about the implementation is in Appendix C.1.

**Active Learning Protocol.** We evaluate all those active learning methods in a batch-mode setup with an initial set size $M = 500$ for all those three data sets, batch size $b$ varying from $\{250, 500, 1000\}$. And, after each query, no re-initialization is used.

## 5.2 RESULTS AND ANALYSIS

The main experimental results have been provided as plots due to the limited space. We also provide tables in which we report the mean and standard deviation for each plot in Appendix E.3.

**Overall Results.** The average test accuracy at each query round is shown in Figure 3. Our method *dynamicAL* can consistently outperform other methods for all query rounds. This suggests that *dynamicAL* is a good choice regardless of the labeling budget. Besides, we notice *dynamicAL* can work on data sets with a large class number, such as Caltech101. However, the baseline method, BADGE, cannot be scaled up to those data sets, because the required memory is linear with the number of classes. Note, *dynamicAL* depends on the quality of pseudo labeling, thus a proper initial labeled set is important. This figure shows that *dynamicAL* is able to work well with a relatively small initial labeled set ($M = 500$). Due to the limited space, we only show the result under three different setting in Figure 3. More evaluation results are in Appendix E.2 and Appendix E.5.

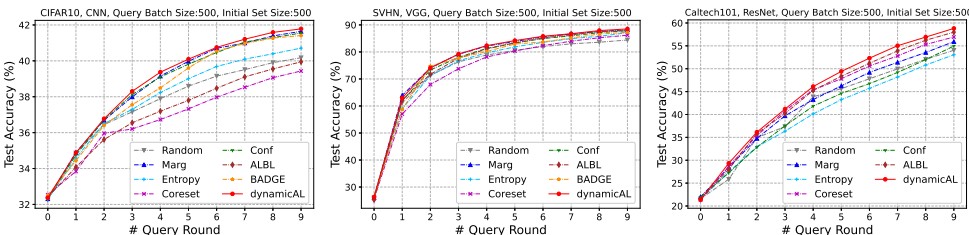

Figure 3: Active learning test accuracy versus the number of query round for a range of conditions.

**Effect of Query Size and Query Round.** Given the total label budget $B$, the increasing of query size always leads to the decreasing of query round. We study the influence of different query size and query round on *dynamicAL* from two perspectives. First, we study the expected approximation ratio with different query batch size on different data sets. As shown in Figure 4, under different settings the expected approximation ratio always converges to 1 with the increase of training epochs, which further indicates that the query set selected by using the approximated change of training dynamics is a reasonably good result for the query set selection problem. Second, we study influence of query round for actual performance of target models. The performance for different target models on different data sets with total budge size $B = 1000$ is shown in Table 1. For certain query budget, our active learning algorithm can be further improved if more query rounds is allowed.

Table 1: Accuracy of *dynamicAL* with different query batch size $b$.

| Method | CIFAR10 + CNN | CIFAR10 + Resnet | SVHN + VGG | Caltech101 + Resnet |
|---|---|---|---|---|
| $R = 10, b = 100$ | **36.84** | **40.92** | **76.34** | **37.06** |
| $R = 4, b = 250$ | 36.72 | 40.78 | 75.26 | 36.48 |
| $R = 2, b = 500$ | 36.71 | 40.46 | 74.10 | 35.91 |
| $R = 1, b = 1000$ | 36.67 | 40.09 | 70.04 | 33.82 |

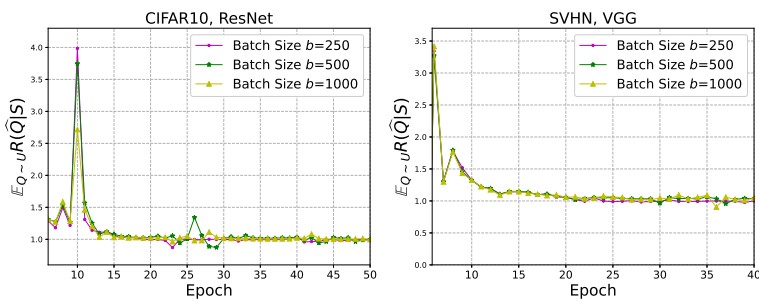

Figure 4: The Expectation of the Approximation Ratio with different query batch size $b$.

**Comparison with Different Variants.** The active learning criterion of *dynamicAL* can be written as $\sum_{(x,y)\in S} \|\nabla_\theta \ell(f(x;\theta_u), \hat{y}_u)\|^2 + \gamma \nabla_\theta \ell(f(x_u;\theta), \hat{y}_u)^\top \nabla_\theta \ell(f(x;\theta), y)$. We empirically show the performance for $\gamma \in \{0, 1, 2, \infty\}$ in Figure 5. As the prediction of our analysis (Equation 15), for $\gamma = 2$, the model achieves the best performance. The result confirms the importance of theoretical analysis for the design of deep active learning methods.

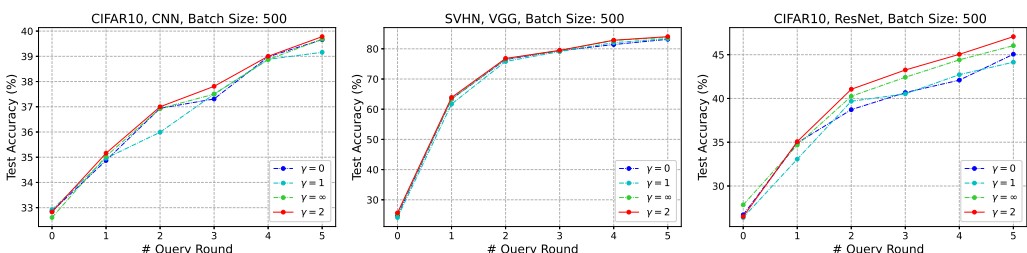

Figure 5: Test Accuracy of different variants.

# 6 RELATED WORK

**Neural Tangent Kernel (NTK):** Recent study has shown that under proper conditions, an infinite-width neural network can be simplified as a linear model with Neural Tangent Kernel (NTK) (Jacot et al., 2018). Since then, NTK has become a powerful theoretical tool to analyze the behavior of deep learning model (CNN, GNN, RNN) from its output dynamics (Arora et al., 2019a; Du et al., 2019; Alemohammad et al., 2020) and to characterize the convergence and generalization error (Arora et al., 2019b). Besides, (Hanin & Nica, 2020) studies the finite-width NTK, aiming at making the NTK more practical.

**Active Learning:** Active learning aims at interactively query labels for unlabeled data points to maximize model performances (Settles, 2009). Among others, there are two popular strategies for active learning, *i.e.*, diversity sampling (Du et al., 2015; Volpi et al., 2018; Zhdanov, 2019), uncertainty sampling (Roy & McCallum, 2001; Zhu & Ma, 2012; Yang & Loog, 2016; Ash et al., 2020; Yoo & Kweon, 2019; Settles et al., 2007; Liu et al., 2021). Recently, there are several papers proposed to used gradient to measure uncertainty (Settles et al., 2007; Ash et al., 2020; Liu et al., 2021). However, those methods need to compute gradient for each class, they can hardly be applied on data sets with a large class number.

# 7 CONCLUSION

In this work, we bridge the gap between the theoretic findings of deep neural networks and real-world deep active learning applications. By exploring the connection between the generalization performance and the training dynamics, we propose a theory-driven method, *dynamicAL*, which selects samples to maximize training dynamics. We prove that the convergence speed of training and the generalization performance is (positively) strongly correlated under the ultra-wide condition and we show that maximizing the training dynamics will lead to a lower generalization error. Empirically, our work show that *dynamicAL* not only consistently outperforms strong baselines across various setting, but also scales well on large deep learning models. We provide the discussion of the limitation of future work in Appendix F.

## 8 REPRODUCIBILITY

To ensure the results and conclusions of our paper are reproducible, we make the following efforts:

Empirically, we provide the URL for the code, and instructions needed to reproduce the main experimental results. And we specify all the training and implementation details in Section 5 and Appendix C.1. Besides, we independently run experiments five times and report the mean and standard deviation in Appendix E.3.
Theoretically, we state the full set of assumptions and include complete proofs of our theoretical results in Section 4 and Appendix B.

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

## A  APPENDIX: DERIVATION OF OBJECTIVES

For the notational convenience, we use $f(x)$ to represent $f(x;\theta)$ in the Appendix.

### A.1  TRAINING DYNAMICS FOR CROSS-ENTROPY LOSS

The partial derivative for softmax function can be defined with the following,

$$\frac{\partial \sigma^i(f(x))}{\partial f^j(x)} = \begin{cases} \sigma^i(f(x))\big(1 - \sigma^i(f(x))\big), & i = j, \\ -\sigma^i(f(x))\sigma^j(f(x)), & i \neq j \end{cases} \tag{24}$$

Then, we have:

$$\begin{aligned}
\frac{\partial \ell(f(x), y)}{\partial t} &= -\sum_i y^i \frac{\partial \log \sigma^i f(x)}{\partial \sigma^i(f(x))} \frac{\partial \sigma^i(f(x))}{\partial t} \\
&= -\sum_i y^i \frac{1}{\sigma^i(f(x))} \sum_j \frac{\partial \sigma^i(f(x))}{\partial f^j(x)} \frac{\partial f^j(x)}{\partial t} \\
&= -\sum_i y^i \sum_j \big(\mathbb{1}[i == j] - \sigma^j(f(x))\big) \frac{\partial f^j(x)}{\partial t} \\
&= -\sum_i \big(y^i - \sigma^i(f(x))\big) \nabla_\theta f^i(x) \nabla_t \theta
\end{aligned} \tag{25}$$

### A.2  DERIVATION FOR CROSS-ENTROPY LOSS

$$\begin{aligned}
\frac{\partial \ell(f(x), y)}{\partial \theta} &= \frac{\partial \ell}{\partial f(x)} \frac{\partial f(x)}{\partial \theta} = -\sum_i y^i \frac{1}{\sigma^i(f(x))} \frac{\partial \sigma^i(f(x))}{f(x)} \frac{\partial f(x)}{\partial \theta} \\
&= -\sum_i y^i \frac{1}{\sigma^i(f(x))} \sigma^i(f(x)) \sum_j \big(\mathbb{1}[i == j] - \sigma^j(f(x))\big) \frac{\partial f^j(x)}{\partial \theta} \\
&= \sum_j \big(\sigma^j(f(x)) - y^j\big) \frac{\partial f^j(x)}{\partial \theta}
\end{aligned} \tag{26}$$

### A.3  APPENDIX: TRAINING DYNAMICS FOR MEAN SQUARED ERROR

For the labeled data set $S$, we define the Mean Squared Error(MSE) as:

$$L_{MSE}(S) = \sum_{(x,y) \in S} \ell_{MSE}(f(x), y) = -\sum_{(x,y) \in S} \sum_{i \in [K]} \frac{1}{2}(f^i(x) - y^i)^2$$

Then the training loss dynamics for each sample can define as:

$$\frac{\partial \ell_{MSE}(f(x), y)}{\partial t} = -\sum_i \big(y^i - f^i(x)\big) \nabla_\theta f^i(x) \nabla_t \theta$$

Because neural networks are optimized by gradient descent, thus:

$$\nabla_t \theta = \theta_{t+1} - \theta_t = \sum_{(x,y) \in S} \frac{\partial \ell(f(x), y)}{\partial \theta} = \sum_{(x,y) \in S} \sum_j \big(f^j(x) - y^j\big) \frac{\partial f^j(x)}{\partial \theta}$$

Therefore, the training dynamics of MSE loss can be expressed as:

$$G_{MSE}(S) = -\frac{1}{\eta} \frac{\partial \sum_{(x,y) \in S} \ell_{MSE}(f(x), y)}{\partial t} = (f(X) - Y)^\top \mathcal{K}(X, X)(f(X) - Y)$$

## A.4 APPENDIX: DECOMPOSITION OF THE CHANGE OF TRAINING DYNAMICS

According to the definition of training dynamics ( Equation (8) ), we have,

$$G(S) = \sum_{i,j} \sum_{(x_l, y_l) \in S} \left( \sigma^i(f(x_l; \theta)) - y_l^i \right) \sum_{(x_{l'}, y_{l'}) \in S} \nabla_\theta f^i(x_l; \theta)^\top \nabla_\theta f^j(x_{l'}; \theta) \left( \sigma^j(f(x_{l'}; \theta)) - y_{l'}^j \right)$$

$$G(S \cup \widehat{Q}) = \sum_{i,j} \sum_{(x, y) \in S \cup \widehat{Q}} \left( \sigma^i(f(x; \theta)) - y^i \right) \sum_{(x', y') \in S \cup \widehat{Q}} \nabla_\theta f^i(x; \theta)^\top \nabla_\theta f^j(x'; \theta) \left( \sigma^j(f(x'; \theta)) - y'^j \right)$$

The change of training dynamics, $\Delta(\widehat{Q}|S) = G(S \cup \widehat{Q}) - G(S)$, can be further simplified as:

$$\begin{aligned}
\Delta(\widehat{Q}|S) &= G(S \cup \widehat{Q}) - G(S) \\
&= 2 \sum_{i,j} \sum_{(x_u, \widehat{y}_u) \in \widehat{Q}} \left( \sigma^i(f(x_u; \theta)) - \widehat{y}_u^i \right) \sum_{(x_l, y_l) \in S} \nabla_\theta f^i(x_u; \theta)^\top \nabla_\theta f^j(x_l; \theta) \left( \sigma^j(f(x_l; \theta)) - y_l^j \right) \\
&+ \sum_{i,j} \sum_{(x_u, \widehat{y}_u) \in \widehat{Q}} \left( \sigma^i(f(x_u; \theta)) - \widehat{y}_u^i \right) \nabla_\theta f^i(x_u; \theta)^\top \nabla_\theta f^j(x_u; \theta) \left( \sigma^j(f(x_u; \theta)) - \widehat{y}_u^j \right) \\
&+ \sum_{i,j} \sum_{(x_u, \widehat{y}_u) \in \widehat{Q}} \left( \sigma^i(f(x_u; \theta)) - \widehat{y}_u^i \right) \sum_{(x_{u'}, \widehat{y}_{u'}) \in \widehat{Q}, u' \neq u} \nabla_\theta f^i(x_{u'}; \theta)^\top \nabla_\theta f^j(x_{u'}; \theta) \left( \sigma^j(f(x_{u'}; \theta)) - \widehat{y}_{u'}^j \right) \\
&= \sum_{(x_u, \widehat{y}_u) \in \widehat{Q}} \Delta(\{(x_u, \widehat{y}_u)\}|S) + \sum_{(x_u, \widehat{y}_u), (x_{u'}, \widehat{y}_{u'}) \in \widehat{Q}} d^i(x_u, \widehat{y}_u)^\top \mathcal{K}^{ij}(x_u, x_{u'}) d^i(x_{u'}, \widehat{y}_{u'})
\end{aligned}$$

## A.5 APPENDIX: SIMPLIFICATION OF THE CHANGE OF TRAINING DYNAMICS

$$\begin{aligned}
\Delta(\{(x_u, \widehat{y}_u)\}|S) =& 2 \sum_{i,j} \sum_{(x_u, \widehat{y}_u) \in \widehat{Q}} \left( \sigma^i(f(x_u; \theta)) - \widehat{y}_u^i \right) \sum_{(x_l, y_l) \in S} \nabla_\theta f^i(x_u; \theta)^\top \nabla_\theta f^j(x_l; \theta) \left( \sigma^j(f(x_l; \theta)) - y_l^j \right) \\
&+ \sum_{i,j} \sum_{(x_u, \widehat{y}_u) \in \widehat{Q}} \left( \sigma^i(f(x_u; \theta)) - \widehat{y}_u^i \right) \nabla_\theta f^i(x_u; \theta)^\top \nabla_\theta f^j(x_u; \theta) \left( \sigma^j(f(x_u; \theta)) - \widehat{y}_u^j \right)
\end{aligned}$$

The derivative of loss with respect to model parameters can be written as:

$$\frac{\partial \sum_{(x, y) \in S} \ell(f(x; \theta), y)}{\partial \theta} = \sum_{(x, y) \in S} \sum_{j \in [K]} \left( \sigma^j(f(x; \theta)) - y^j \right) \nabla_\theta f^j(x; \theta)$$

Therefore, the change of training dynamics caused by $\{(x_u, \widehat{y}_u)\}$ can be written as:

$$\Delta(\{(x_u, \widehat{y}_u)\}|S) = \|\nabla_\theta \ell(f(x_u; \theta), \hat{y}_u)\|^2 + 2 \sum_{(x, y) \in S} \nabla_\theta \ell(f(x_u; \theta), \hat{y}_u)^\top \nabla_\theta \ell(f(x; \theta), y)$$

# B APPENDIX: PROOFS FOR THEORETICAL ANALYSIS

## B.1 PROOFS FOR THEOREM 1

**Lemma 1** (Convergence Analysis with NTK, Theorem 4.1 of Arora et al. (2019b)). *Suppose $\lambda_0 = \lambda_{\min}(\Theta) > 0$ for all subsets of data samples. For $\delta \in (0, 1)$, if $m = \Omega(\frac{n^7}{\lambda_0^4 \delta^4 \epsilon^2})$ and $\eta = O(\frac{\lambda_0}{n^2})$. Then with probability at least $1 - \delta$, the network can achieve near-zero training error,*

$$\|Y - f_t(X; \theta(t))\|_2 = \sqrt{\sum_{k=1}^{K} \sum_{i=1}^{n} (1 - \eta \lambda_i)^{2t} (\vec{v}_i^\top Y^k)^2} \pm \epsilon \tag{27}$$

*where $n$ denotes the number of training samples, $m$ the width of hidden layers. The NTK $\Theta = V^\top \Lambda V$ with $\Lambda = \{\lambda_i\}_{i=1}^n$ a diagonal matrix of eigenvalues and $V = \{\vec{v}_i\}_{i=1}^n$ a unitary matrix.*

*Proof.* According to Arora et al. (2019b), if $m = \Omega(\frac{n^7}{\lambda_0^4 \delta^4 \epsilon^2})$ and learning ratio $\eta = O(\frac{\lambda_0}{n^2})$, then with probability at least $1 - \delta$ over the random initialization, we have, $\|Y_l - f_t(X; \theta(t))\|_2 = \sqrt{\sum_{k=1}^K \sum_{i=1}^n (1 - \eta\lambda_i)^{2t} (v_i^\top Y_l^k)^2} \pm \epsilon$. We decompose the NTK using $\boldsymbol{\Theta} = V^\top \Lambda V$ with $\Lambda = \{\lambda_i\}_{i=1}^n$ a diagonal matrix of eigenvalues and $V = \{v_i\}_{i=1}^n$ a unitary matrix. At each training step in active learning, the labeled samples will be updated by $S = S \cup \overline{Q}$. We can apply the convergence result in each of this step and achieve near zero error. $\qquad\square$

**Theorem 1** (Relationship between convergence rate and alignment)**.** *Under the same assumptions as in Lemma 1. The convergence rate described by $\mathcal{E}_t$ satisfies,*

$$\mathrm{Tr}[Y^\top Y] - 2t\eta\mathcal{A}(X,Y) \leq \mathcal{E}_t^2(X,Y) \leq \mathrm{Tr}[Y^\top Y] - \eta\mathcal{A}(X,Y) \tag{28}$$

*Proof.* We first prove the inequality on the right hand side. It is easy to see that $(1-\eta\lambda_i)^{2t} \leq (1-\eta\lambda_i)$ for each $\lambda_i$ and $t \geq 1$, based on the fact that $\forall\lambda_i, 0 \leq 1 - \eta\lambda_i \leq 1$. Then we can obtain,

$$\mathcal{E}_t(X,Y) = \sqrt{\sum_{k=1}^K \sum_{i=1}^n (1 - \eta\lambda_i)^{2t} (v_i^\top Y^k)^2} \leq \sqrt{\sum_{k=1}^K \sum_{i=1}^n (1 - \eta\lambda_i)(v_i^\top Y^k)^2}$$

$$= \sqrt{\mathrm{Tr}[Y^\top(I - \eta\boldsymbol{\Theta})Y]} = \sqrt{\mathrm{Tr}[Y^\top Y] - \eta\mathcal{A}(X,Y)}$$

Then we use Bernoulli's inequality to prove the inequality on the left hand side. Bernoulli's inequality states that, $(1 + x)^r \geq 1 + rx$, for every integer $r \geq 0$ and every real number $x \geq -1$. It is easy to check that $(-\eta\lambda_i) \geq -1, \forall\lambda_i$. Therefore,

$$\mathcal{E}_t(X,Y) = \sqrt{\sum_{k=1}^K \sum_{i=1}^n (1 - \eta\lambda_i)^{2t} (v_i^\top Y^k)^2} \geq \sqrt{\sum_{k=1}^K \sum_{i=1}^n (1 - 2t\eta\lambda_i)(v_i^\top Y^k)^2}$$

$$= \sqrt{\mathrm{Tr}[Y^\top(I - 2t\eta\boldsymbol{\Theta})Y]} = \sqrt{\mathrm{Tr}[Y^\top Y] - 2t\eta\mathcal{A}(X,Y)}$$

$$\square$$

## B.2 PROOF FOR THEOREM 2

**Lemma 2** (Generalization bound with NTK, Theorem 5.1 of Arora et al. (2019b))**.** *Suppose data $S = \{(x_i, y_i)\}_{i=1}^n$ are i.i.d. samples from a non-degenerate distribution $p(x,y)$, and $m \geq \mathrm{poly}(n, \lambda_0^{-1}, \delta^{-1})$. Consider any loss function $\ell : \mathbb{R} \times \mathbb{R} \to [0,1]$ that is 1-Lipschitz. Then with probability at least $1 - \delta$ over the random initialization, the network trained by gradient descent for $T \geq \Omega(\frac{1}{\eta\lambda_0}\log\frac{n}{\delta})$ iterations has population risk $\mathcal{L}_p = \mathbb{E}_{(x,y)\sim p(x,y)}[\ell(f_T(x), y)]$ that is bounded as follows:*

$$\mathcal{L}_p \leq \sqrt{\frac{2\,\mathrm{Tr}[Y^\top\boldsymbol{\Theta}^{-1}(X,X)Y]}{n}} + O\left(\sqrt{\frac{\log\frac{n}{\lambda_0\delta}}{n}}\right). \tag{29}$$

*Proof.* We first show that the generalization bound regrading our method on ultra-wide networks. The distance between weights of trained networks and their initialization values can be bounded as, $\|w_r(t) - w_r(0)\| = O(\frac{n}{\sqrt{m}\lambda_0\sqrt{\delta}})$. We then give a bound on the $\|W(t) - W(0)\|_F$, where $W = \{w_1, w_2, \dots\}$ is the set of all parameters. We definite $Z = \frac{\partial f(t)}{\partial W(t)}$, then the update function is given by $W(t+1) = W(t) - \eta Z(Z^\top W(t) - Y)$. Summing over all the time step $t = 0, 1, \dots$, we can obtain that $W(\infty) - W(0) = \sum_{t=0}^\infty \eta Z(I - \eta\boldsymbol{\Theta})y = Z\boldsymbol{\Theta}^{-1}Y$. Thus the distance can be measured by $\|W(\infty) - W(0)\|_F^2 = \mathrm{Tr}[Y^\top\boldsymbol{\Theta}^{-1}Y]$.

Then the key step is to apply Rademacher complexity. Given $R > 0$, with probability at least $1 - \delta$, simultaneously for every $B > 0$, the function class $\mathcal{F}_{B,R} = \{f : \|w_r(t) - w_r(0)\| \leq R \ (\forall r \in m), \|W(\infty) - W(0)\|_F^2 \leq B\}$ has empirical Rademacher complexity bounded as,

$$\mathcal{R}_S(\mathcal{F}_{B,R}) = \frac{1}{n}\mathbb{E}_{\epsilon_i\in\{\pm 1\}^n}\left[\sup_{f\in\mathcal{F}_{B,R}}\sum_{i=1}^n \epsilon_i f(x_i)\right] \leq \frac{B}{\sqrt{2n}}\left(1 + (\frac{2\log\frac{2}{\delta}}{m})^{1/4}\right) + 2R^2\sqrt{m} + R\sqrt{2\log\frac{2}{\delta}}$$

where $B = \sqrt{\text{Tr}[Y^\top \boldsymbol{\Theta}^{-1}(X,X)Y]}$, and $R = \frac{n}{\sqrt{m}\lambda_0\sqrt{\delta}}$.

Finally, Rademacher complexity directly gives an upper bound on generalization error (Mohri et al., 2018), $\sup_{f\in\mathcal{F}}\{\mathcal{L}_p(f) - \mathcal{L}_S(f)\} \leq 2\mathcal{R}_S + 3c\sqrt{\frac{\log(2/\delta)}{2n}}$, where $\mathcal{L}_S(f) \leq \frac{1}{\sqrt{n}}$. Based on this, we apply a union bound over a finite set of different $i$'s. Then with probability at least $1 - \delta/3$ over the sample $S$, we have $\sup_{f\in\mathcal{F}_{R,B_i}}\{\mathcal{L}_p(f) - \mathcal{L}_S(f)\} \leq 2\mathcal{R}_S(\mathcal{F}_{B_i,R}) + O(\sqrt{\frac{\log\frac{n}{\lambda_0\delta}}{n}})$, $\forall i \in \{1, 2, \ldots, O(\frac{n}{\lambda_0})\}$. Taking a union bound, we know that with probability at least $1 - \frac{2}{3}\delta$ over the sample $S$, we have, $f_T \in \mathcal{F}_{B_i^*,R}$ for some $i^*$, $\mathcal{R}_S(\mathcal{F}_{B_i^*,R}) \leq \sqrt{\frac{\text{Tr}[Y^\top\boldsymbol{\Theta}^{-1}(X,X)Y]}{2n}} + \frac{2}{\sqrt{n}}$ and $\sup_{f_T\in\mathcal{F}_{B_i^*,R}}\{\mathcal{L}_p(f_T) - \mathcal{L}_S(f_T)\} \leq 2\mathcal{R}_S(\mathcal{F}_{B_i^*,R}) + O(\sqrt{\frac{\log\frac{n}{\lambda_0\delta}}{n}})$. These together can imply,

$$\mathcal{L}_p(f) \leq \frac{1}{\sqrt{n}} + 2\mathcal{R}_S(\mathcal{F}_{B_i^*,R}) + O(\sqrt{\frac{\log\frac{n}{\lambda_0\delta}}{n}}) \leq \sqrt{\frac{2\,\text{Tr}[Y^\top\boldsymbol{\Theta}^{-1}(X,X)Y]}{n}} + O\left(\sqrt{\frac{\log\frac{n}{\lambda_0\delta}}{n}}\right).$$

More proof details can be found in Arora et al. (2019b). $\qquad\square$

**Theorem 2** (Relationship between the generalization bound and alignment). *Under the same assumptions as in Lemma* (2). *If we define the generalization upper bound as $\mathcal{B}(X,Y) = \sqrt{\frac{2\,\text{Tr}[Y^\top\boldsymbol{\Theta}^{-1}Y]}{n}}$, then it can be bounded with the alignment as follows,*

$$\frac{\text{Tr}^2[Y^\top Y]}{\mathcal{A}(X,Y)} \leq \frac{n}{2}\mathcal{B}^2(X,Y) \leq \frac{\lambda_{max}}{\lambda_{min}}\frac{\text{Tr}^2[Y^\top Y]}{\mathcal{A}(X,Y)} \tag{30}$$

*Proof.* We first expand the following expression:

$$\frac{n}{2}\mathcal{B}^2(X,Y)\mathcal{A}(X,Y) = \sum_{k=1}^K\sum_{i=1}^n \lambda_i(v_i^\top Y^k)^2 \sum_{k=1}^K\sum_{i=1}^n \frac{1}{\lambda_i}(v_i^\top Y^k)^2$$

Then we use this expansion to prove the inequality on the left hand side,

$$\sum_{k=1}^K\sum_{i=1}^n \lambda_i(v_i^\top Y^k)^2 \sum_{k=1}^K\sum_{i=1}^n \frac{1}{\lambda_i}(v_i^\top Y^k)^2 = \sum_{k=1}^K\sum_{k'=1}^K\left(\sum_{i=1}^n \lambda_i(v_i^\top Y^k)^2 \sum_{i=1}^n \frac{1}{\lambda_i}(v_i^\top Y^{k'})^2\right)$$

$$\geq \sum_{k=1}^K\sum_{k'=1}^K\left(\sum_{i=1}^n(v_i^\top Y^k)^2 \sum_{i=1}^n(v_i^\top Y^{k'})^2\right) = \left(\sum_{k=1}^K Y^{k\top}V^\top VY^k\right)\left(\sum_{k=1}^K Y^{k\top}V^\top VY^k\right)$$

$$= \text{Tr}^2[Y^\top Y]$$

The second line is due to quadratic mean is greater or equal to geometric mean. Finally, we prove the inequality on the right hand side,

$$\sum_{k=1}^K\sum_{i=1}^n \lambda_i(v_i^\top Y^k)^2 \sum_{k=1}^K\sum_{i=1}^n \frac{1}{\lambda_i}(v_i^\top Y^k)^2 = \sum_{k=1}^K\sum_{k'=1}^K\left(\sum_{i=1}^n \lambda_i(v_i^\top Y^k)^2 \sum_{i=1}^n \frac{1}{\lambda_i}(v_i^\top Y^{k'})^2\right)$$

$$\leq \sum_{k=1}^K\sum_{k'=1}^K \frac{\lambda_{max}}{\lambda_{min}}\left(\sum_{i=1}^n(v_i^\top Y^k)^2 \sum_{i=1}^n(v_i^\top Y^{k'})^2\right) = \frac{\lambda_{max}}{\lambda_{min}}\left(\sum_{k=1}^K Y^{k\top}V^\top VY^k\right)\left(\sum_{k=1}^K Y^{k\top}V^\top VY^k\right)$$

$$= \frac{\lambda_{max}}{\lambda_{min}}\,\text{Tr}^2[Y^\top Y]$$

$\qquad\square$

### B.3 DERIVATION FOR MAXIMUM MEAN DISCREPANCY

The difference between truth risk over $p(x)$ and $q(x)$ can be defined as,

$$\mathcal{L}_p - \mathcal{L}_q = \int_x g(x)p(x)dx - \int_x g(x)q(x)dx$$

where $g(x) = \int_y \ell(f(x;\theta),y)p(y|x)dy$. Follow Wang & Ye (2015), we assume that the prediction functions have bounded norm $\|f\|_F$. Thus, the function $g$ is bounded. By given the loss function, $g$ is also measurable. Then, $\exists \hat{g} \in \mathcal{C}(x)$, such that,

$$\int_x g(x)p(x)dx - \int_x g(x)q(x)dx = \int_x \hat{g}(x)p(x)dx - \int_x \hat{g}(x)q(x)dx$$

$$\leq \sup_{\hat{g}\in\mathcal{C}(x)} \int_x \hat{g}(x)p(x)dx - \int_x \hat{g}(x)q(x)dx = \text{MMD}\big(p(x),q(x),\mathcal{C}\big)$$

where $\mathcal{C}(x)$ is the function class of bounded and continuous functions of $x$. To make the MMD term be measurable, we empirically restrict the MMD on a reproducing kernel Hilbert space (RKHS) with the characteristic kernel $\mathcal{H}_{\Theta}$. Following Gretton et al. (2012), we know that the relationship between the true MMD and the empirical MMD is,

$$P\Big(\big|\text{MMD}\big(p(x),q(x),\mathcal{C}\big) - \text{MMD}(S_0,S,\mathcal{H}_{\Theta})\big| \geq \epsilon + 2(\sqrt{\frac{C}{n_0}} + \sqrt{\frac{C}{n}})\Big) \leq 2e^{\frac{-\epsilon^2 n_0 n}{2C(n_0+n)}}$$

where $\text{MMD}(S_0,S,\mathcal{H}_{\Theta})$ is the empirical measure for $\text{MMD}(p(x),q(x),\mathcal{H}_{\Theta})$. Slightly overloading the notation, we denote $S \sim q(x)$, which may not be i.i.d., and the initial label set $S_0 \sim p(x)$. Then, in the active learning setting, $S_0 \subseteq S$. Further, we denote $|S_0| = n_0, |S| = n$ and $\forall x, x' \in S, \Theta(x,x') \leq C$. Therefore, we have, $\sqrt{\frac{C}{n}} + \sqrt{\frac{C}{n_0}} \geq 2\sqrt{\frac{C}{n}}$. For constant factor $\gamma = \frac{M}{M+B}$, we have the following inequality,

$$P\big(\text{MMD}\big(p(x),q(x),\mathcal{C}\big) \geq \text{MMD}(S_0,S,\mathcal{H}_{\Theta}) + \epsilon + 4\sqrt{\frac{C}{n}}\big) \leq 2e^{\frac{-\gamma\epsilon^2 n}{4C}}$$

Denoting $2e^{\frac{-\gamma\epsilon^2 n}{4C}} = \delta/2$, then we have $\epsilon = \sqrt{\frac{4C\ln(4/\delta)}{\gamma n}}$. Combining all the above results, we show that with probability at least $1 - \delta$, the following inequality holds:

$$\mathcal{L}_p - \mathcal{L}_q \leq \text{MMD}(S_0,S,\mathcal{H}_{\Theta}) + 4\sqrt{\frac{C}{n}} + \sqrt{\frac{4C\ln(4/\delta)}{\gamma n}}$$

Then, we can get,

$$\mathcal{L}_p - \mathcal{L}_q \leq \text{MMD}(S_0,S,\mathcal{H}_{\Theta}) + O\left(\sqrt{\frac{C\ln(1/\delta)}{n}}\right)$$

## C APPENDIX: MORE DETAILS OF EXPERIMENTAL SETTINGS

### C.1 IMPLEMENTATION DETAIL

For simple CNN model, we utilize the same architecture used in Pytorch CIFAR10 Image Classification Tutorial [1]. For ResNet model, we use the Pytorch Offical implementation of ResNet-18 [2] and set the output dimension to the number of classes. For VGG model, we use the Pytorch Offical implementation of VGG-11 [3]. Besides, we leverage the library BackPACK (Dangel et al., 2020) to collect the gradient of samples in batch.

---

[1] https://pytorch.org/tutorials/beginner/blitz/cifar10_tutorial.html
[2] https://github.com/pytorch/vision/blob/main/torchvision/models/resnet.py
[3] https://github.com/pytorch/vision/blob/main/torchvision/models/vgg.py

We keep a constant learning rate of 0.001 for all three datasets and all three models. All the codes mentioned above use the MIT license. All experiments are done with four Tesla V100 SXM2 GPUs and a 12-core 2.2GHz CPU.

## D APPENDIX: VERIFICATION EXPERIMENTS UNDER ULTRA-WIDE CONDITION

### D.1 EXPERIMENT SETTING AND COMPUTATIONAL DETAIL FOR THE EMPIRICAL COMPARISON BETWEEN NTK AND MMD

**Experiment Setting** For the verification experiment shown in Figure 1, we employ a simple CNN as the target model, in which there are three convolutional layers following with global average pooling layer, on the CFAIR10 data set. Note, this CNN architecture is widely used in NTK analysis works (Arora et al., 2019a; Zandieh et al., 2021). To make the verification experiment close to the application setting, we keep size of initial labeled set and query batch size same as what we used in Section 5.

**Computational Detail** We follow (Jia et al., 2021) to compute the MMD with NTK kernel. The MMD term, $\text{MMD}(p(x), q(x), \mathcal{H}_\Theta)$, can be simplified into the following form:

$$\text{MMD}^2(p(x), q(x)) = \mathbb{E}[\boldsymbol{\Theta}(x_i, x_j) + \boldsymbol{\Theta}(x_i', x_j') - 2\boldsymbol{\Theta}(x_i, x_j')] \tag{31}$$

where $x_i, x_j \sim p(x)$ and $x_i', x_j' \sim q(x)$. Then, we define set $S_0$ as $\{x_1, ..., x_{n_0}\} \sim p(x)$ and set $S$ as $\{x_1', ..., x_n'\} \sim q(x)$, where $n_0 \leq n$. The $\text{MMD}^2(S_0, S)$ is an unbiased estimation for $\text{MMD}^2(p(x), q(x))$, can we explicitly computed by:

$$
\begin{aligned}
\text{MMD}^2(S_0, S) &= \frac{1}{m^2 - m}a + \frac{1}{n^2 - n}b - \frac{2}{m(n-1)}c \\
a &= \left( \sum_{i,j}^m \boldsymbol{\Theta}(x_i, x_j) - \sum_i^m \boldsymbol{\Theta}(x_i, x_i) \right) \\
b &= \left( \sum_{i,j}^n \boldsymbol{\Theta}(x_i', x_j') - \sum_i^n \boldsymbol{\Theta}(x_i', x_i') \right) \\
c &= \left( \sum_i^m \sum_j^n \boldsymbol{\Theta}(x_i, x_j') - \sum_{i,j}^m \boldsymbol{\Theta}(x_i, x_i') \right)
\end{aligned}
\tag{32}
$$

Therefore, the MMD term of Equation (23), $\text{MMD}(S_0, S, \mathcal{H}_\Theta)$, can be empirically approximated by $\sqrt{\text{MMD}^2(S_0, S)}$.

### D.2 EXPERIMENT SETTING FOR THE CORRELATION STUDY BETWEEN TRAINING DYNAMICS AND ALIGNMENT

**Experiment Setting** For the verification experiment shown in Figure 2, we also use the simple CNN on CIFAR10. And to keep consistent with the application setting, we set $|S| = 500$ and $|\overline{Q}| = 250$. The $\overline{Q}$ is randomly sampled from the unlabeled set and the labeled set $S$ is fixed. We independently sample $\overline{Q}$ 150 times to compute the correlation between between $G_{MSE}(S \cup \overline{Q})$ and $\mathcal{A}(X\|X_Q, Y\|Y_Q)$.

## E APPENDIX: MORE DETAILS OF EXPERIMENTAL RESULTS

### E.1 BASELINES

1. Random: Unlabeled data are randomly selected at each round.

2. Coreset: This method performs a clustering over the last hidden representations in the network, and calculates the minimum distance between each candidate sample's embedding

and embeddings of labeled samples. Then data samples with the maximum distances are selected. (Sener & Savarese, 2018).

3. Confidence Sampling (Conf): The method selects $b$ examples with smallest predicted class probability $\max_i^K f^i(x; \theta)$ (Wang & Shang, 2014).

4. Margin Sampling (Marg): The bottom $b$ examples sorted according to the example's multi-class margin are selected. The margin is defined as $f^i(x; \theta) - f^j(x; \theta)$, where $i$ and $j$ are the indices of the largest and second largest entries of $f(x; \theta)$ (Roth & Small, 2006).

5. Entropy: Top $b$ samples are selected according to the entropy of the example's predictive class probability distribution, the entropy is defined as $H((f^i(x; \theta))_{i=1}^K)$, where $H(p) = \sum_i^K p_i \ln \frac{1}{p_i}$ (Wang & Shang, 2014).

6. Active Learning by Learning (ALBL): The bandit-style meta-active learning algorithm combines Coreset and Conf (Hsu & Lin, 2015).

7. Batch Active learning by Diverse Gradient Embeddings (BADGE): $b$ samples are selected by using k-means++ seeding on the gradients of the final layer, in order to query by uncertainty and diversity. (Ash et al., 2020).

## E.2 EXPERIMENT RESULTS

The results for ResNet18, VGG11, and vanilla CNN on CIFAR10, SVHN, and Caltech101 with different batch sizes have been shown in the Figure 3 and 6. Note, for the large batch size setting ($b = 1000$) on Caltech101, we set the number of query round $R = 4$, in which 49.2% images will be labeled after 4 rounds.

## E.3 NUMERICAL RESULT OF MAIN EXPERIMENTS

For the the main experiments, we report the means and standard deviations of active learning performance under different settings in the the following tables.

Table 2: CIFAR10, CNN, Query Batch Size:250, Initial Set Size:500

|   | Random | Marg | Entropy | Coreset | Conf | ALBL | BADGE | *dynamicAL* |
|---|--------|------|---------|---------|------|------|-------|-------------|
| 0 | 32.32%±1.308% | 32.48%±1.286% | 32.32%±1.269% | 32.41%±1.281% | 32.27%±1.266% | 32.50%±1.263% | 32.66%±1.182% | 32.57%±1.423% |
| 1 | 33.00%±1.175% | 33.16%±1.165% | 32.74%±1.617% | 32.75%±1.306% | 33.00%±1.703% | 32.98%±1.184% | 33.45%±1.813% | 33.52%±1.311% |
| 2 | 34.14%±1.322% | 34.41%±1.130% | 34.06%±1.546% | 33.77%±1.011% | 34.21%±1.426% | 34.02%±1.392% | 34.66%±1.483% | 34.70%±1.019% |
| 3 | 35.05%±1.508% | 35.50%±1.301% | 35.16%±1.679% | 34.44%±0.937% | 35.25%±1.344% | 34.97%±1.227% | 35.75%±1.024% | 35.78%±1.115% |
| 4 | 35.64%±1.945% | 36.55%±1.249% | 36.14%±1.646% | 35.08%±1.396% | 36.59%±1.508% | 35.58%±1.177% | 36.33%±0.791% | 36.72%±0.716% |
| 5 | 36.28%±1.124% | 37.18%±1.547% | 36.77%±1.004% | 35.68%±1.390% | 37.19%±1.063% | 36.15%±1.311% | 37.29%±1.126% | 37.45%±1.573% |
| 6 | 36.88%±1.568% | 37.73%±1.546% | 37.28%±1.983% | 36.18%±1.419% | 37.65%±2.062% | 36.65%±1.111% | 37.90%±1.988% | 37.95%±1.414% |
| 7 | 37.29%±1.605% | 38.01%±0.874% | 37.67%±1.723% | 36.57%±1.346% | 38.09%±1.174% | 37.07%±1.731% | 38.28%±1.474% | 38.41%±1.295% |
| 8 | 37.59%±1.848% | 38.43%±1.675% | 38.01%±1.601% | 36.98%±0.748% | 38.58%±1.556% | 37.35%±1.135% | 38.51%±1.091% | 38.70%±1.291% |
| 9 | 37.85%±1.789% | 38.75%±1.550% | 38.29%±1.312% | 37.25%±1.527% | 38.91%±1.902% | 37.57%±1.170% | 38.78%±0.776% | 38.91%±1.358% |

Table 3: CIFAR10, CNN, Query Batch Size:500, Initial Set Size:500

|   | Random | Marg | Entropy | Coreset | Conf | ALBL | BADGE | *dynamicAL* |
|---|--------|------|---------|---------|------|------|-------|-------------|
| 0 | 32.26%±1.164% | 32.31%±1.441% | 32.29%±1.397% | 32.54%±1.331% | 32.32%±1.288% | 32.41%±1.432% | 32.49%±1.320% | 32.37%±1.049% |
| 1 | 34.87%±1.286% | 34.89%±1.575% | 34.58%±1.664% | 33.84%±1.368% | 34.75%±1.503% | 34.08%±1.368% | 34.44%±1.230% | 34.88%±1.557% |
| 2 | 36.45%±0.842% | 36.69%±1.456% | 36.50%±1.463% | 35.96%±1.667% | 36.73%±1.744% | 35.62%±1.536% | 36.41%±1.175% | 36.78%±1.253% |
| 3 | 37.16%±0.767% | 37.99%±1.356% | 37.30%±1.221% | 36.20%±1.086% | 38.12%±1.663% | 36.55%±1.327% | 37.56%±1.284% | 38.30%±1.152% |
| 4 | 37.89%±0.880% | 39.15%±1.056% | 38.23%±0.878% | 36.73%±1.011% | 39.10%±1.336% | 37.20%±1.381% | 38.49%±1.238% | 39.37%±0.708% |
| 5 | 38.59%±0.861% | 39.98%±1.562% | 39.01%±1.278% | 37.33%±1.373% | 39.81%±1.402% | 37.80%±1.560% | 39.61%±1.219% | 40.09%±0.940% |
| 6 | 39.15%±1.108% | 40.70%±1.391% | 39.68%±1.315% | 37.97%±1.393% | 40.47%±1.126% | 38.47%±1.270% | 40.55%±1.066% | 40.75%±1.671% |
| 7 | 39.51%±1.219% | 40.99%±1.217% | 40.09%±1.408% | 38.53%±1.600% | 41.05%±1.448% | 39.11%±1.385% | 40.97%±0.814% | 41.21%±1.433% |
| 8 | 39.90%±0.807% | 41.39%±1.614% | 40.39%±1.357% | 39.06%±1.156% | 41.30%±1.865% | 39.55%±1.595% | 41.27%±1.409% | 41.59%±1.013% |
| 9 | 40.17%±1.170% | 41.64%±1.287% | 40.71%±0.739% | 39.43%±0.892% | 41.55%±1.341% | 39.95%±1.299% | 41.41%±0.949% | 41.78%±0.645% |

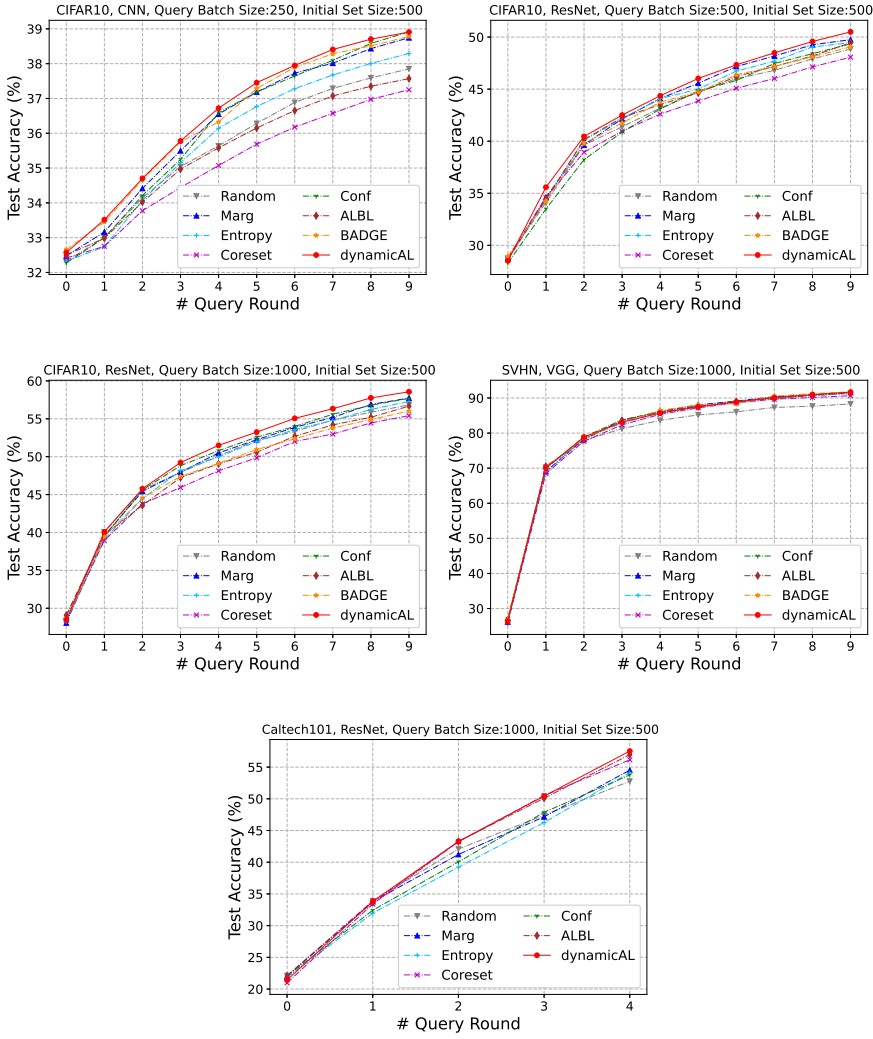

Figure 6: The evaluation results for different active learning methods under a range of conditions.

Table 4: CIFAR10, ResNet, Query Batch Size:500, Initial Set Size:500

|   | Random | Marg | Entropy | Coreset | Conf | ALBL | BADGE | dynamicAL |
|---|---|---|---|---|---|---|---|---|
| 0 | 28.75%±1.780% | 28.75%±0.369% | 28.63%±1.394% | 28.63%±1.120% | 28.31%±1.011% | 28.75%±0.957% | 28.95%±1.040% | 28.52%±0.686% |
| 1 | 34.11%±3.088% | 34.62%±2.022% | 34.42%±0.849% | 34.57%±0.992% | 33.49%±1.269% | 34.42%±2.077% | 34.26%±1.740% | 35.58%±2.858% |
| 2 | 39.63%±2.157% | 39.63%±0.313% | 40.08%±1.022% | 38.94%±1.408% | 38.23%±1.454% | 40.16%±2.574% | 39.78%±1.384% | 40.46%±0.959% |
| 3 | 41.38%±2.357% | 42.15%±0.810% | 42.18%±1.271% | 40.96%±0.961% | 40.87%±0.860% | 42.26%±2.347% | 41.74%±1.230% | 42.51%±0.799% |
| 4 | 43.18%±1.809% | 44.09%±1.165% | 44.09%±1.150% | 42.60%±1.094% | 43.10%±1.325% | 43.52%±3.064% | 43.76%±1.364% | 44.36%±0.980% |
| 5 | 44.73%±2.253% | 45.57%±1.115% | 45.00%±0.731% | 43.86%±1.369% | 44.83%±1.388% | 44.64%±3.097% | 44.73%±1.675% | 46.02%±0.754% |
| 6 | 46.00%±2.193% | 47.17%±0.929% | 46.74%±1.118% | 45.08%±1.549% | 45.83%±1.426% | 46.22%±2.601% | 46.38%±1.607% | 47.34%±1.027% |
| 7 | 46.80%±2.134% | 48.18%±1.230% | 47.69%±1.253% | 46.02%±1.589% | 47.47%±1.424% | 47.18%±2.384% | 47.17%±1.404% | 48.48%±1.452% |
| 8 | 47.91%±1.722% | 49.26%±0.652% | 49.05%±1.113% | 47.14%±1.880% | 48.40%±1.178% | 48.18%±2.503% | 48.11%±1.049% | 49.58%±1.673% |
| 9 | 48.84%±1.584% | 49.75%±1.341% | 49.46%±1.282% | 48.07%±1.480% | 49.35%±1.269% | 49.45%±2.529% | 49.06%±0.850% | 50.50%±1.301% |

Table 5: CIFAR10, ResNet, Query Batch Size:1000, Initial Set Size:500

| | Random | Marg | Entropy | Coreset | Conf | ALBL | BADGE | *dynamicAL* |
|---|---|---|---|---|---|---|---|---|
| 0 | 28.34%±1.465% | 28.07%±2.604% | 28.24%±1.756% | 28.41%±0.722% | 29.05%±1.137% | 29.06%±0.847% | 28.43%±1.176% | 28.48%±2.062% |
| 1 | 40.08%±0.329% | 39.57%±1.551% | 39.09%±2.180% | 38.95%±1.047% | 39.50%±2.340% | 39.67%±1.489% | 39.46%±3.020% | 40.09%±1.795% |
| 2 | 45.63%±1.253% | 45.43%±0.444% | 44.48%±1.823% | 43.78%±0.986% | 45.62%±1.882% | 43.58%±1.329% | 44.55%±3.654% | 45.77%±2.290% |
| 3 | 47.90%±1.257% | 47.96%±0.735% | 48.15%±2.509% | 45.93%±0.682% | 48.82%±1.797% | 47.24%±1.926% | 47.39%±4.189% | 49.22%±1.704% |
| 4 | 50.13%±1.207% | 50.49%±0.807% | 49.97%±2.819% | 48.14%±0.566% | 50.79%±1.870% | 49.05%±1.831% | 49.13%±4.053% | 51.50%±1.925% |
| 5 | 52.14%±1.517% | 52.24%±0.781% | 52.00%±2.762% | 49.85%±1.075% | 52.59%±2.202% | 50.59%±1.636% | 50.94%±3.628% | 53.24%±1.927% |
| 6 | 53.33%±1.300% | 53.87%±0.635% | 53.57%±3.123% | 52.01%±0.772% | 53.99%±2.390% | 52.69%±1.599% | 52.36%±3.924% | 55.06%±1.697% |
| 7 | 54.84%±1.238% | 55.19%±1.136% | 54.79%±3.144% | 52.99%±1.147% | 55.60%±2.002% | 54.20%±1.685% | 53.77%±3.985% | 56.33%±1.613% |
| 8 | 55.86%±1.161% | 56.90%±0.732% | 56.23%±3.182% | 54.45%±0.821% | 56.79%±2.033% | 55.20%±1.868% | 54.91%±4.104% | 57.76%±1.796% |
| 9 | 56.84%±0.979% | 57.73%±0.500% | 57.29%±3.225% | 55.42%±0.954% | 57.70%±2.042% | 56.67%±1.783% | 56.02%±3.935% | 58.56%±1.574% |

Table 6: SVHN, VGG, Query Batch Size:500, Initial Set Size:500

| | Random | Marg | Entropy | Coreset | Conf | ALBL | BADGE | *dynamicAL* |
|---|---|---|---|---|---|---|---|---|
| 0 | 25.94%±7.158% | 26.15%±6.290% | 26.41%±8.994% | 25.83%±5.845% | 26.52%±7.489% | 25.31%±5.030% | 26.38%±9.100% | 26.30%±5.505% |
| 1 | 61.23%±4.812% | 63.93%±3.127% | 59.02%±4.724% | 57.02%±3.672% | 61.99%±2.613% | 62.14%±5.531% | 58.70%±5.615% | 63.01%±12.293% |
| 2 | 71.35%±2.364% | 74.08%±0.933% | 71.25%±1.459% | 67.95%±2.870% | 73.31%±2.828% | 71.75%±2.555% | 74.74%±2.978% | 74.10%±4.557% |
| 3 | 76.34%±1.626% | 79.17%±1.064% | 76.74%±1.521% | 73.76%±2.844% | 78.02%±1.939% | 77.65%±1.518% | 77.75%±2.100% | 79.20%±2.651% |
| 4 | 78.86%±1.378% | 82.18%±0.504% | 79.67%±0.809% | 78.14%±2.486% | 81.32%±1.901% | 81.09%±1.005% | 80.16%±1.353% | 82.33%±2.134% |
| 5 | 80.56%±1.149% | 83.85%±0.750% | 81.87%±0.638% | 80.34%±2.339% | 83.31%±1.529% | 83.37%±1.225% | 82.94%±0.830% | 84.19%±1.940% |
| 6 | 81.98%±1.334% | 85.61%±0.624% | 83.56%±0.541% | 82.32%±1.592% | 84.94%±0.858% | 85.19%±0.993% | 83.69%±0.975% | 85.80%±1.498% |
| 7 | 83.00%±1.048% | 86.62%±0.607% | 84.94%±0.079% | 83.98%±1.394% | 85.97%±1.179% | 86.31%±0.977% | 85.15%±0.760% | 86.75%±1.426% |
| 8 | 83.59%±0.945% | 87.57%±0.625% | 85.78%±0.068% | 85.26%±1.431% | 87.13%±0.679% | 87.55%±0.831% | 86.61%±0.478% | 87.91%±1.264% |
| 9 | 84.42%±0.744% | 88.23%±0.600% | 87.11%±0.437% | 86.18%±0.886% | 87.87%±0.598% | 87.89%±0.780% | 87.29%±0.441% | 88.52%±1.240% |

Table 7: SVHN, VGG, Query Batch Size:1000, Initial Set Size:500

| | Random | Marg | Entropy | Coreset | Conf | ALBL | BADGE | *dynamicAL* |
|---|---|---|---|---|---|---|---|---|
| 0 | 25.90%±3.479% | 26.20%±5.409% | 26.85%±4.403% | 26.18%±6.853% | 27.21%±8.721% | 26.60%±4.688% | 26.88%±6.248% | 26.43%±8.047% |
| 1 | 70.26%±3.154% | 69.06%±3.646% | 68.72%±2.156% | 68.46%±1.111% | 69.85%±3.485% | 70.51%±3.487% | 70.09%±2.690% | 70.04%±1.650% |
| 2 | 77.91%±1.061% | 78.24%±2.237% | 78.56%±0.492% | 77.66%±1.784% | 78.89%±2.809% | 78.14%±1.494% | 78.67%±1.799% | 78.86%±1.710% |
| 3 | 81.25%±0.812% | 83.68%±1.657% | 82.83%±0.527% | 82.34%±1.461% | 83.75%±2.165% | 83.50%±1.669% | 83.07%±1.334% | 83.11%±1.269% |
| 4 | 83.63%±0.746% | 86.12%±1.251% | 85.80%±0.744% | 85.34%±1.126% | 85.91%±1.128% | 86.18%±0.979% | 86.50%±1.087% | 85.70%±1.179% |
| 5 | 85.17%±0.870% | 88.04%±1.022% | 87.66%±0.683% | 87.19%±0.928% | 87.61%±1.044% | 87.65%±1.031% | 88.03%±0.742% | 87.46%±1.054% |
| 6 | 86.06%±0.822% | 89.13%±0.712% | 88.96%±0.395% | 88.65%±0.505% | 88.90%±0.845% | 88.93%±0.809% | 88.41%±0.783% | 88.89%±1.274% |
| 7 | 87.30%±0.948% | 90.36%±0.532% | 90.00%±0.257% | 89.65%±0.486% | 90.18%±0.706% | 89.83%±0.747% | 90.53%±0.495% | 90.09%±1.149% |
| 8 | 87.69%±0.890% | 90.95%±0.375% | 90.67%±0.385% | 90.15%±0.410% | 90.96%±0.677% | 90.75%±0.567% | 91.25%±0.432% | 90.95%±0.782% |
| 9 | 88.28%±0.723% | 91.59%±0.417% | 91.25%±0.353% | 90.64%±0.311% | 91.66%±0.755% | 91.41%±0.665% | 91.76%±0.367% | 91.67%±0.840% |

Table 8: Caltech101, ResNet, Query Batch Size:500, Initial Set Size:500

| | Random | Marg | Entropy | Coreset | Conf | ALBL | *dynamicAL* |
|---|---|---|---|---|---|---|---|
| 0 | 21.59%±1.431% | 21.98%±1.688% | 21.49%±1.681% | 21.39%±1.738% | 21.98%±1.459% | 21.48%±1.828% | 21.38%±1.323% |
| 1 | 25.84%±1.112% | 28.42%±1.677% | 27.43%±0.760% | 28.61%±1.224% | 27.25%±1.155% | 28.02%±1.515% | 29.34%±2.111% |
| 2 | 34.94%±0.635% | 34.76%±1.745% | 32.94%±1.224% | 35.37%±1.561% | 32.85%±0.849% | 35.86%±1.012% | 36.14%±1.234% |
| 3 | 37.34%±1.088% | 39.70%±1.328% | 36.36%±0.636% | 40.81%±1.005% | 37.52%±1.250% | 40.20%±1.091% | 41.19%±0.789% |
| 4 | 43.87%±0.867% | 43.26%±0.612% | 40.12%±0.805% | 45.38%±0.508% | 41.82%±1.104% | 45.27%±0.725% | 46.11%±1.138% |
| 5 | 45.45%±1.672% | 46.25%±1.562% | 43.24%±1.617% | 47.81%±1.683% | 44.60%±1.295% | 48.35%±1.729% | 49.42%±1.298% |
| 6 | 47.60%±1.383% | 49.20%±1.310% | 45.71%±1.047% | 50.60%±1.596% | 46.74%±0.760% | 51.20%±1.466% | 52.31%±1.739% |
| 7 | 49.97%±0.530% | 51.40%±1.571% | 48.19%±0.928% | 52.80%±1.887% | 49.19%±0.885% | 53.90%±1.166% | 55.03%±1.098% |
| 8 | 52.06%±1.476% | 53.56%±1.044% | 50.81%±0.943% | 55.31%±1.105% | 51.99%±1.383% | 56.22%±0.838% | 56.92%±1.153% |
| 9 | 54.04%±0.898% | 55.92%±0.496% | 53.05%±0.554% | 56.93%±0.691% | 54.96%±0.981% | 57.99%±0.805% | 58.81%±1.040% |

Table 9: Caltech101, ResNet, Query Batch Size:1000, Initial Set Size:500

| | Random | Marg | Entropy | Coreset | Conf | ALBL | *dynamicAL* |
|---|---|---|---|---|---|---|---|
| 0 | 22.13%±1.050% | 22.05%±1.011% | 21.83%±0.725% | 20.98%±0.631% | 22.03%±1.364% | 22.05%±0.633% | 21.42%±1.735% |
| 1 | 33.91%±1.330% | 33.80%±1.002% | 31.98%±1.000% | 33.40%±0.962% | 32.43%±0.895% | 33.66%±2.174% | 33.83%±1.438% |
| 2 | 42.08%±0.560% | 41.22%±0.730% | 39.23%±0.981% | 43.24%±0.960% | 40.05%±0.988% | 43.28%±2.360% | 43.27%±2.280% |
| 3 | 47.43%±0.700% | 47.16%±0.659% | 46.26%±0.968% | 50.51%±0.706% | 47.87%±0.698% | 50.10%±2.082% | 50.43%±1.634% |
| 4 | 52.77%±0.980% | 54.52%±1.288% | 54.11%±1.347% | 56.15%±1.284% | 53.76%±1.196% | 56.96%±1.733% | 57.52%±1.189% |

### E.4 MAXIMUM MEAN DISCREPANCY FOR MULTIPLE ROUNDS

As shown in Figure 1, the MMD term is much smaller than the $\mathcal{B}$ at the first query round. To better understand the relation between MMD and $\mathcal{B}$ for multiple query setting, we measure the MMD/$\mathcal{B}$

for $R \geq 2$. As shown in Figure 7, $\mathcal{B}$ is much larger than MMD even multiple query rounds. Besides, we notice that, for the first round, the larger query batch always leads to larger MMD$/\mathcal{B}$, because the sampling bias introduced by the query policy will be amplified by using large batch size.

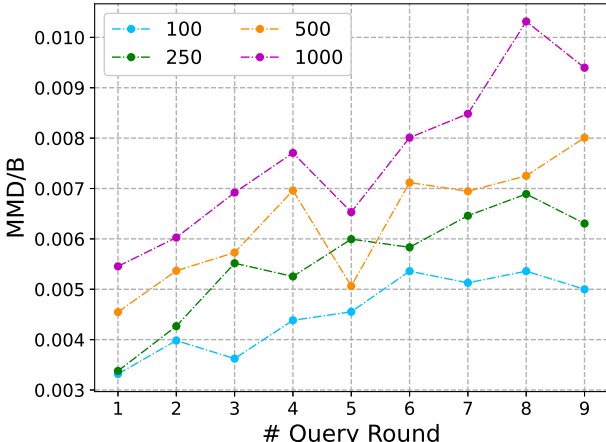

Figure 7: MMD$/\mathcal{B}$ for larger query round.

Furthermore, we measure the MMD$/\mathcal{B}$ with a constant total budget size but different query rounds. The result is shown in Table 10. As our expectation, spending the total query budget in one query round will induce the largest MMD$/\mathcal{B}$. And, with more query rounds, the MMD$/\mathcal{B}$ will be lower.

Table 10: MMD$/\mathcal{B}$ under constant budget size.

| | $R = 10, b = 100$ | $R = 4, b = 250$ | $R = 2, b = 500$ | $R = 1, b = 1000$ |
|---|---|---|---|---|
| MMD$/\mathcal{B}$ | 0.004999 | 0.005253 | 0.005367 | 0.005455 |

### E.5 EXPERIMENT RESULT IN RETRAINING SETTING

In Section 5, we mainly focus on the non-retraining setting, in which the weights of the network obtained in the previous will be used for the next AL cycle. To further study the effectiveness of *dynamicAL* in the retraining setting, in which the network will be reinitialized after each query, we compare *dynamicAL* with the strong baseline in the retraining active learning setting, such as Coreset (Sener & Savarese, 2018). We follow Ash et al. (2020) to query samples when training accuracy be greater than 99% and the results are shown in Table 11 and 12. The results show that *dynamicAL* can still be better than or competitive with the commonly used active learning methods. Besides, we notice that the improvement in the non-retraining setting is more significant, because the dynamic analysis (Equation (8)) considers the change of dynamics according to the model's current parameters.

Table 11: CIFAR10, ResNet, Query Batch Size 500, Initial Set Size 500.

| #Round | Random | Coreset | *dynamicAL* |
|---|---|---|---|
| 0 | 30.80±1.81 | 30.77±0.92 | 30.94±2.17 |
| 1 | 35.80±1.52 | 36.62±2.10 | 36.47±0.13 |
| 2 | 42.91±1.75 | 43.16±1.79 | 42.74±2.44 |
| 3 | 43.76±0.65 | 44.35±2.25 | 46.43±1.07 |
| 4 | 47.03±1.19 | 48.74±1.94 | 49.38±1.80 |
| 5 | 49.16±1.77 | 50.20±1.25 | 51.61±1.09 |
| 6 | 52.43±1.33 | 53.44±1.37 | 54.33±1.76 |
| 7 | 52.81±1.55 | 53.89±0.78 | 54.59±1.04 |
| 8 | 54.56±0.23 | 57.12±1.11 | 57.50±1.28 |
| 9 | 58.08±1.48 | 59.62±1.50 | 60.35±1.80 |

Table 12: SVHN, VGG, Query Batch Size 500, Initial Set Size 500.

| #Round | Random | Coreset | *dynamicAL* |
|---|---|---|---|
| 0 | 52.68±1.97 | 52.74±6.16 | 52.59±3.73 |
| 1 | 67.64±1.99 | 68.08±3.61 | 66.48±4.10 |
| 2 | 73.46±1.51 | 74.93±1.44 | 74.34±2.22 |
| 3 | 77.30±1.08 | 76.49±2.08 | 76.73±2.65 |
| 4 | 79.27±0.78 | 79.33±0.72 | 80.19±0.78 |
| 5 | 79.97±1.28 | 82.09±1.08 | 82.08±1.39 |
| 6 | 83.97±0.42 | 82.30±0.33 | 83.80±1.30 |
| 7 | 83.44±0.57 | 83.29±1.11 | 84.85±1.12 |
| 8 | 86.24±0.52 | 84.72±0.52 | 86.59±1.25 |
| 9 | 85.75±1.23 | 85.62±0.55 | 86.57±0.74 |

## F    LIMITATION AND FUTURE WORK

Our proposed active learning method, *dynamicAL*, is designed for the small budget, non-retraining setting in active learning, which is commonly seen in practice, and tends to be more challenging than other settings (Mittal et al., 2019). However, as shown in Appendix E.5, if retraining from scratch is affordable in a certain application, the performance of our method is comparable with state-of-the-art techniques, such as Coreset (Sener & Savarese, 2018). Moreover, there are some works trying to combine the techniques of self-supervised and semi-supervised learning with active learning (Mittal et al., 2019; Chan et al., 2021). As demonstrated in the recent semi-supervised learning works (Sohn et al., 2020; Rizve et al., 2021), the data augmentation, consistency regularization and uncertainty-aware pseudo-label selection will boost the quality of the pseudo labeling that *dynamicAL* relies on. Then, potentially, they can improve the performance of *dynamicAL*. Besides, large initial set size and properly selected initial set (Yuan et al., 2011) also potentially improve the quality of the pseudo labeling. Therefore, future research directions include theoretically analyzing the effect of semi-supervised learning technologies for active learning and empirically verifying the effectiveness of combining *dynamicAL* and semi-supervised learning technologies in retraining and non-retraining active learning settings.

