# OpenReview forum: "Deep Active Learning by Leveraging Training Dynamics"
_ICLR.cc/2022/Conference — ICLR 2022 Submitted_

### Official Review · Reviewer_j1Xn · 2021-10-31

**Correctness:** 4
**Technical Novelty And Significance:** 3
**Empirical Novelty And Significance:** 2
**Recommendation:** 6
**Confidence:** 3

**Main Review:**

Strength:
The paper is technically clear and well-written, with good result on toy benchmarks (i.e., clean, small images with mostly balanced distribution). The method seems to be a good continuation of Ash et al., 2020; Liu et al., 2021 and built a theoretical bridge between these objectives with the training dynamics and generalization in deep learning theory justification.

Weakness:
* The infinite-width assumption: The main theoretical result (Theorem 1) is derived under the assumption of infinite width model. However, it is well known that the infinite assumption does not hold in practice (e.g., [Fort et al, 2020](https://arxiv.org/abs/2010.15110)), which has led the community to derive better approximations such as NTH ([Huang and Yau, 2019](https://arxiv.org/abs/1909.08156)) or their more tangible lower-order approximations ([Chen et al, 2020](https://proceedings.neurips.cc/paper/2020/file/b6b90237b3ebd1e462a5d11dbc5c4dae-Paper.pdf)). While I understand the authors are building their results based on classic NTK theory, it is still worth addressing (can be non-rigorous or based on small empirically verifications) cases where the infinite-width assumption does not hold. Mostly if the high-level conclusion of the theory (faster training -> better generalization) still hold?

* The Pseudo Labeling approach: While I understand this is a technique that is being used in the previous works, I wonder what would  would happen if model prediction is wrong, such that the training dynamics is computed against the wrong label? Some comments about this in either Section 3 or in the conclusion section would be helpful.

* Clarity: This is a minor comment: To make sure the deposition is maximally accessible, please provide derivation for equation (13) and (15) (e.g., in the appendix).

**Summary Of The Paper:**

This work proposes DynamicsAL, a novel AL criteria that selects new training example base on its ability to maximize the training dynamics $\frac{\partial}{\partial t} l(f, y)$.
The authors supplied a practical algorithm (Section 3.2) and compared the proposed criteria with existing methods (Section 3.3).
The authors also derived theoretical result on faster converges leads to better generalization (Section 4) under the assumption that the network has infinite width, and also empirically verified their result based in the non-i.i.d. case. Experiments on standard benchmark and architecture (CNNs on CIFAR/SVHN/Caltech101) are conducted, where DynamicsAL is shown to outperform existing approaches.

**Summary Of The Review:**

A paper that continues the investigation of active learning using gradient/influence function signal (Ash et al., 2020; Liu et al., 2021). It builts a theoretical bridge between these objectives with the concepts of training dynamics and generalization, and supplied a algorithm that is practically effective. I personally find this work to be a nice value add to the community, both in terms of the theoretical contribution and the practical utility of the algorithm. Hence recommend accept.

---

> ### Author Response · Authors · 2021-11-17
> **Author Response to Reviewer j1Xn**
>
> Thank you for your constructive comments!
>
> ### 1. The infinite-width assumption
>
> “It is still worth addressing (can be non-rigorous or based on small empirically verifications) cases where the infinite-width assumption does not hold. Mostly if the high-level conclusion of the theory (faster training -> better generalization) still hold?”
>
> * We acknowledge that some works are discussing the limitations of infinite NTK in the practical setting. However, other recent works have shown the effectiveness of theoretical analysis derived based on infinite NTK for the commonly used models. Some of their applications can achieve SOTA. For example,
>
>   * Park et al. [1] used the NTK to predict the generalization performance of architectures in the application of Neural Architecture Search (NAS).
>   * Chen et al. [2] used the condition number of NTK to predict a model’s trainability.
>   * Chen et al. [3] also used the NTK to evaluate the trainability of several ImageNet models, such as ResNet.
>   * Deshpande et al. [4] used the NTK for model selection in the fine-tuning of pre-trained models on a target task.
>
>   Besides, our experiments on vanilla CNN in Figure 3 and Appendix.E (as suggested “non-rigorous or based on small empirical verifications”) also show the effectiveness of the high-level conclusions derived from the theory still hold.
>
> * It is still worth studying the high-level conclusions for some very deep and very narrow cases from the theoretical perspective. However, as shown in [1][2][3][4] and further verified in our experiments, the key conclusions (faster training -> better generalization) derived based on NTK are still valid and valuable for the commonly used models.
>
> &nbsp;
>
> ### 2. The Pseudo Labeling approach
>
> “I wonder what would happen if model prediction is wrong, such that the training dynamics is computed against the wrong label? Some comments about this in either Section 3 or in the conclusion section would be helpful.”
>
> * Thanks for your great suggestion. We have added more discussion in Section 5.2. Because the quality of pseudo labeling depends on the experiment setting, it will be better to put the discussion in the experiment section.
> * If the model prediction is wrong, the estimation of dynamics will be inaccurate. Then, the several rounds of active learning at the beginning will act as the warm-up. In general, a large initial label set will make a high-quality pseudo labeling and then boost the overall performance. Thus, a proper initial labeled set is important. And in experiments, we show that M=500 is enough for dynamicAL to work well.
>
> &nbsp;
>
> ### 3. Please provide derivation for equations (13) and (15)
>
> * Thanks for your suggestion. We have updated the revision. The derivations of Equations (13) and (15) can be found in Appendix A.4 and A.5.
>
> &nbsp;
>
> ### Reference:
>
> [1] Daniel S Park, Jaehoon Lee, Daiyi Peng, Yuan Cao, and Jascha Sohl-Dickstein. Towards nngp guided neural architecture search. arXiv preprint arXiv:2011.06006, 2020.
>
> [2] Wuyang Chen, Xinyu Gong, and Zhangyang Wang. Neural architecture search on imagenet in four gpu hours: A theoretically inspired perspective. In International Conference on Learning Representations, 2021a.
>
> [3] Chen, Xiangning, Cho-Jui Hsieh, and Boqing Gong. "When Vision Transformers Outperform ResNets without Pretraining or Strong Data Augmentations." arXiv preprint arXiv:2106.01548 (2021).
>
> [4] Deshpande, A., Achille, A., Ravichandran, A., Li, H., Zancato, L., Fowlkes, C., Bhotika, R., Soatto, S. and Perona, P., 2021. A linearized framework and a new benchmark for model selection for fine-tuning. arXiv preprint arXiv:2102.00084.

---

### Official Review · Reviewer_YgGb · 2021-11-01

**Correctness:** 3
**Technical Novelty And Significance:** 4
**Empirical Novelty And Significance:** 1
**Recommendation:** 3
**Confidence:** 3

**Main Review:**

This paper proposes a new active learning strategy that samples examples which would help the network converge faster during training.
The paper suggests that faster convergence during training typically leads to better generalization performance. Therefore it should be beneficial for and AL algorithm to select examples that will speed up training convergence.
This speed-up is measured by approximating the derivative of the of the loss with respect to the training iteration t (which they call training dynamics). The authors propose many relaxations to be able to compute this derivative in practice, notably: (i) when computing the loss, assign weak labels to unlabeled samples, and (ii) instead of computing the derivative on all possible subsets, compute it separately for each unlabeled example then approximate the training dynamics of a subset using the sum of the training dynamics of each sample in that subset.

Strengths:

1. The paper is well written and well organized.
2. The idea of linking training dynamic to active learning is novel and potentially influential.

Weaknesses:
The main weakness of this paper (and the reason I am not recommending it for acceptance) is its experimental section. The theoretical section is well written, but the problem is completely non-tractable and cannot be theoretically analyzed in a realistic setting so the experimental section is instrumental and necessary to support the paper's claims. However, I don't think the experimental section is thorough at all.

1. Coresets AL is a very strong baseline on Cifar10 with a ResNet18, so it is a warning sign that in your experiments it underperforms compared to Random. If you look at Fig 4 in the coresets paper [1], all  AL methods (including Random sampling) hover between 60% and 65% accuracy on Cifar10 with budget 5000 using a VGG-16. I have also run many experiments in that exact setting using a ResNet18 instead and it's easy to reproduce these accuracies.
In fact, figure 3)a) in [2] (which you cite in your paper), runs the same CIFAR10 experiments and reports around 78% for random sampling using 5k samples. However, in the middle-left and top-right plots of your Figure 6 of Appendix E.2, the best accuracy on CIFAR10 when the budget is 5000 using a ResNet18 is below 55%. [2] provides the exact training hyperparameters to reproduce their results so you can try copying them and rerunning your experiments.
The only difference between your experiments and those in the CoreSets paper is that you don't retrain your models from scratch at each round and instead you continue training from the previous checkpoint. So either (a) not resetting the network weights makes a big difference or (b) your training hyperparameters do not allow the network to train to saturation. I don't think (a) is the culprit here because if you look at figure 3)a) in [2] they report 45% on CIFAR10 using 1k randomly selected images while your number is less than 30% (this is when you train from scratch at round 0).
If (b) is true then those plots are inconclusive, since you're not training to saturation and your method is specifically targeting examples that allow faster convergence.
I know that it can be very hard to find good hyperparameters for these experiments because you're trying to use the same learning rate schedule at round 0 as you are on round 9, but the learning rate schedule that works with 1000 training examples is likely less than optimal when training with 9000 examples.
I also understand that it saves computation time and cost to continue training where you left off at the previous round but running at least one experiment where you train from scratch at each round is again necessary to at least show that this isn't the reason your method stands out. Also these experiments can take less than one day when using the machine you mention in appendix C.1.
 Obviously in my comments above I focus on ResNet18 on CIFAR10 because I am familiar with this setting and could find apples to apples comparisons to point you to in [2] , but you should revisit the rest of your experiments on other datasets to verify that they do not suffer from the same problems.

2. I understand that you use small budgets in your experiments to save compute. However, using a small initial budget benefits methods that don't rely heavily on the classifier's feature extractor. If you choose a small initial budget, the feature extractor will be messed up and again methods such as Coresets will completely mess up.


[1] Ozan Sener and Silvio Savarese. Active learning for convolutional neural networks: A core-set approach. In International Conference on Learning Representations, 2018.

[2] Zhuoming Liu, Hao Ding, Huaping Zhong, Weijia Li, Jifeng Dai, and Conghui He. Influence selection for active learning. arXiv preprint arXiv:2108.09331, 2021.

**Summary Of The Paper:**

This paper suggests a new method to perform active learning by choosing prioritizing examples that lead to faster convergence during training.

**Summary Of The Review:**

I found the paper's experimental section to be insufficient.

---

> ### Author Response · Authors · 2021-11-17
> **Author Response to Reviewer YgGb**
>
> Thank you for your constructive comments!
>
> ### 1. The main weakness is its experimental section.
>
> “Coreset AL is a very strong baseline, it is a warning sign that in your experiments it underperforms compared to Random.”
>
> * We would like to make some factual clarification. The effectiveness of Coreset AL has been shown in both [1] and [2] with *retraining* in each query round. Instead, as described in Section 5.1 and Appendix E.2, we continually train the model after the queried samples are obtained (as pointed out by the reviewer). In such *non-retraining* setting, Coreset AL cannot consistently outperform the random method (the response about the training saturation will be provided in the next question). These results provide an empirical sign reminding us that more attention should be paid to the non-retraining setting, which is important because retraining the large model is not always affordable in some real-world applications [3], and the previous methods include Coreset may become ineffective in this setting.
> * Even with retraining, for Coreset+ResNet, Coreset AL cannot always outperform random sampling across different settings. As shown in Figure 12 of [5], with initial label set M=100, query batch b=100, and b=1000, Coreset cannot outperform the random method.
> * “Coreset AL is a very strong baseline” is true when labeled data is sufficient. For both papers pointed out by the reviewer, a large initial label set is used. Specifically, in [1] M=5000 (10% of CF10) and in [2] M=1000.
>
>
> “The only difference between your experiments and those in the CoreSets paper is that you don't retrain your models from scratch at each round and instead you continue training from the previous checkpoint. So either (a) not resetting the network weights makes a big difference or (b) your training hyperparameters do not allow the network to train to saturation. I don't think (a) is the culprit here because if you look at figure 3)a) in [2] they report 45% on CIFAR10 using 1k randomly selected images while your number is less than 30% (this is when you train from scratch at round 0).”
>
> * As we explained above, *the lower performance of Coreset is indeed caused by (a) not resetting the network weights makes a big difference.*
>
> * We would like to point out that it would be unfair to compare our result (30% at round 0) and the result in [2] (45%), because we only used half the amount of data (M=500, described in Section 5.1) used by [2]. Therefore, a lower performance should be expected.
>
> * To further verify that the hyperparameters used in our experiment can make the network train to saturation. We keep the hyperparameters reported in Appendix C but use 1k samples. The target model (ResNet 18) can also achieve 44.3% at R=0 on CF10. Therefore, instead of (b), we believe (a) - resetting the network weights makes a big difference is the reason for the relatively low performance of Resnet on CF10.
>
> * Moreover, we would like to emphasize that our empirical evaluation is fair for all the methods, e.g., the same learning rate, batch size, initial label set size, and stopping criterion. And our method can consistently outperform baseline methods in those settings.

---

> > ### Author Response · Authors · 2021-11-17
> > **Author Response to Reviewer YgGb (Cont.)**
> >
> >
> >
> > “running at least one experiment where you train from scratch at each round is again necessary to at least show that this isn't the reason your method stands out.”
> >
> > * We run two additional experiments in the retraining setting per the reviewer’s suggestion. Our method can still be better than or competitive with the commonly used active learning methods, such as Coreset. And for the non-retraining setting, as shown in Figure 3 and Appendix E, dynamicAL significantly outperforms the others, because the dynamic analysis is for the current model (and current parameters). In all, the effectiveness of dynamicAL under different settings has been demonstrated.
> >
> > &nbsp;
> > **CF10+ResNet, M=500, b=500, lr=0.005**
> >
> > |      | random | coreset | dynamicAL |
> > |----|------|-------|---------|
> > |0|30.8&plusmn;1.81|30.77&plusmn;0.92|30.94&plusmn;2.17|
> > |1|35.8&plusmn;1.52|36.62&plusmn;2.1|36.47&plusmn;0.13|
> > |2|42.91&plusmn;1.75|43.16&plusmn;1.79|42.74&plusmn;2.44|
> > |3|43.76&plusmn;0.65|44.35&plusmn;2.25|46.43&plusmn;1.07|
> > |4|47.03&plusmn;1.19|48.74&plusmn;1.94|49.38&plusmn;1.8|
> > |5|49.16&plusmn;1.77|50.2&plusmn;1.25|51.61&plusmn;1.09|
> > |6|52.43&plusmn;1.33|53.44&plusmn;1.37|54.33&plusmn;1.76|
> > |7|52.81&plusmn;1.55|53.89&plusmn;0.78|54.59&plusmn;1.04|
> > |8|54.56&plusmn;0.23|57.12&plusmn;1.11|57.5&plusmn;1.28|
> > |9|58.08&plusmn;1.48|59.62&plusmn;1.5|60.35&plusmn;1.8|
> >
> >
> > &nbsp;
> > **SVHN+VGG, M=500, b=500, lr=0.001**
> >
> > |      | random | coreset | dynamicAL |
> > | ---- | ------ | ------- | --------- |
> > |0|52.68&plusmn;1.97|52.74&plusmn;6.16|52.59&plusmn;3.73|
> > |1|67.64&plusmn;1.99|68.08&plusmn;3.61|66.48&plusmn;4.1|
> > |2|73.46&plusmn;1.51|74.93&plusmn;1.44|74.34&plusmn;2.22|
> > |3|77.3&plusmn;1.08|76.49&plusmn;2.08|76.73&plusmn;2.65|
> > |4|79.27&plusmn;0.78|79.33&plusmn;0.72|80.19&plusmn;0.78|
> > |5|79.97&plusmn;1.28|82.09&plusmn;1.08|82.08&plusmn;1.39|
> > |6|83.97&plusmn;0.42|82.3&plusmn;0.33|83.8&plusmn;1.3|
> > |7|83.44&plusmn;0.57|83.29&plusmn;1.11|84.85&plusmn;1.12|
> > |8|86.24&plusmn;0.52|84.72&plusmn;0.52|86.59&plusmn;1.25|
> > |9|85.75&plusmn;1.23|85.62&plusmn;0.55|86.57&plusmn;0.74|
> >
> >
> > &nbsp;
> >
> >
> > ###  2. If you choose a small initial budget, the feature extractor will be messed up and again methods such as Coreset will completely mess up.
> >
> > * We agree with the reviewer that, if the feature extractor is trained on a small initial set, the representative power might be limited. Consequently, some methods, such as Coreset which highly depends on the representation for distance computation, will have poor performance. However, the lack of labeled data is a commonly encountered problem in real-world active learning scenarios. Besides, some recent works even (e.g., [4]) start to research the without-initial-annotation setting (i.e., the initial set is empty). Therefore, in this paper, we focus on a relatively small initial training set in our experiments.
> >
> > &nbsp;
> >
> > ### Reference:
> >
> > [1] Ozan Sener and Silvio Savarese. Active learning for convolutional neural networks: A core-set approach. In International Conference on Learning Representations, 2018.
> >
> > [2] Zhuoming Liu, Hao Ding, Huaping Zhong, Weijia Li, Jifeng Dai, and Conghui He. Influence selection for active learning. arXiv preprint arXiv:2108.09331, 2021.
> >
> > [3] Ostapuk, Natalia, Jie Yang, and Philippe Cudré-Mauroux. "Activelink: deep active learning for link prediction in knowledge graphs." The World Wide Web Conference. 2019.
> >
> > [4] Liu, Zhao-Yang, and Sheng-Jun Huang. "Active sampling for open-set classification without initial annotation." Proceedings of the AAAI Conference on Artificial Intelligence. Vol. 33. No. 01. 2019.
> >
> > [5] Jordan T. Ash, Chicheng Zhang, Akshay Krishnamurthy, John Langford, and Alekh Agarwal. Deep batch active learning by diverse, uncertain gradient lower bounds. InInternational Conference on Learning Representations, 2020.

---

> > > ### Comment · Reviewer_YgGb · 2021-11-18
> > > **Reply to Rebuttal**
> > >
> > > Thank you for you for you thorough response.  Below are some more questions I have:
> > >
> > > 1. Thank you for clarifying that performance differences are indeed caused by non-retraining. I actually stumbled upon another ICLR submission that makes the same remark [1], which is very interesting. This should be handled with care in the paper by explicitly mentioning that performance boosts are less significant in the retraining setting (or simply including the new tables your provided in your response in a plot).
> > > 3. In the **CF10+ResNet, M=500, b=500, lr=0.005** table you provide in your response, the entry in the second row (index 1 in your table), first column (random), corresponds to training a randomly initialized network on 1000 randomly sampled images (M=500 at round 0 + b=500 at round 1)? If so, then shouldn't the reported 35.8% accuracy be closer to 44.3% as you explain in your comment `To further verify that the hyperparameters used in our experiment can make the network train to saturation. We keep the hyperparameters reported in Appendix C but use 1k samples. The target model (ResNet 18) can also achieve 44.3% at R=0 on CF10.` Please clarify this point in case I am misreading the table.
> > > 4. `We would like to point out that it would be unfair to compare our result (30% at round 0) and the result in [2] (45%), because we only used half the amount of data (M=500, described in Section 5.1) used by [2]. ` Please remind the reader in the figure caption or on the x-axis what the initial budget was.
> > > 5. `Besides, some recent works even (e.g., [4]) start to research the without-initial-annotation setting (i.e., the initial set is empty). Therefore, in this paper, we focus on a relatively small initial training set in our experiments.` That is a valid setting, but in that case, you have tools at your disposal (self-supervised/semi-supervised learning and data augmentation for example) that can significantly boost model performance and as mentioned in [1], [2], and [3] some of these tools can make some successful AL algorithms obsolete compared to random sampling. For example, in figure 6 in [2], the authors show that integrating SSL with AL in the low budget regime makes all baselines underperform compared to random sampling. If the paper is focused on the small initial training set because it's more realistic, then I am still curious about the performance with SSL for example, this will boost the quality of your algorithm's pseudo-labels which it relies on, but is that boost sufficient to beat random sampling?
> > > Also the main benefit of AL is to reduce annotation costs. These costs are only significant in the large scale settings (ImageNet) or on challenging tasks (segmentation), so what we call "low budget" regime on CIFAR-10 is very unrealistic anyway: even if you assume it takes 10 seconds to manually label a single image, it would take a single person around 80 minutes to label the 500 images in your initial set. In realistic settings, there are now a number of companies dedicated to data labeling alone and they employ hundreds if not thousands of human annotators.
> > >
> > > I am still reluctant to give the paper a higher score because of the experimental section. The AL benchmarking framework has come under scrutiny in [2], and [3] so I would expect new works to consider/comment on all the factors mentioned in these papers. I would improve my score if the authors clearly emphasize the limitations of the experimental section: small budget, small datasets, non-retraining, no self/semi-supervision, no augmentation, etc. and at least comment on the potential positive or negative impacts that each of these factors could have on the proposed algorithm.
> > >
> > >
> > > [1] Best Practices in Pool-based Active Learning for Image Classification, Submitted to The Tenth International Conference on Learning Representations 2022, https://openreview.net/forum?id=7Rnf1F7rQhR, under review
> > >
> > > [2] Mittal, Sudhanshu et al. “Parting with Illusions about Deep Active Learning.” ArXiv abs/1912.05361 (2019): n. pag.
> > >
> > > [3] Yao-Chun Chan, Mingchen Li, andSamet Oymak, On the Marginal Benefit of Active Learning: Does Self-Supervision Eat Its Cake?

---

> > > > ### Author Response · Authors · 2021-11-20
> > > > **Thank you for your suggestions**
> > > >
> > > > Thank you for your suggestions! We hope our answers below address your concerns.
> > > >
> > > >
> > > > ### 1. Mention that performance boosts are less significant in the retraining setting
> > > >
> > > > * We will include the two tables in Appendix E.5 and add the “Limitation and Future Work” discussion section in Appendix F, in which the performance in the retraining setting will be discussed.
> > > >
> > > > &nbsp;
> > > >
> > > > ###  2. Shouldn't the reported 35.8% accuracy be closer to 44.3%
> > > >
> > > > * In retraining experiments, we didn’t use the query strategy (when to query) of [1]. In their work, querying new samples at which epoch is manually selected. Although elaborately selecting query epoch can make the ResNet achieve 45% under their hyperparameters and 44.3% under our hyperparameters, we don’t think that way is general enough because there is typically no validation set in the active learning setup, and selecting a good query epoch may need to rely on the test set. Note that, in a real-world scenario, for a new data set, we cannot access the test labels and there is no parameter tuning experience that can be borrowed. Instead, in the retraining setting (two tables presented above), we follow the query strategy of [2] and allow active learning methods to query samples when the training accuracy is greater than 0.99 (Line 63 of [3]). Therefore, it is unfair to directly compare the results achieved in [1] and the results with the query strategy of [2]. The purpose of the retraining experiments is to respond to the following comment: “show that this (non-retraining) isn't the (only) reason your method stands out.”
> > > > * As we mentioned in the first point, we have included the retraining experiments and provide the detail of query strategy in Appendix E.5.
> > > >
> > > > &nbsp;
> > > >
> > > > ### 3. Remind the reader in the figure caption or on the x-axis what the initial budget was.
> > > >
> > > > * Thanks for your suggestion. We have included the initial budget information in both the figures (Figure 3 & 6) and table captions (Table 2-9).
> > > >
> > > > &nbsp;
> > > >
> > > > ### 4. More advanced tools in your setting
> > > >
> > > > “I am still curious about the performance with SSL for example, this will boost the quality of your algorithm's pseudo-labels which it relies on, but is that boost sufficient to beat random sampling?”
> > > >
> > > > * We acknowledge that borrowing some techniques from other research fields, such as data augmentation and consistency regularization can further boost the performance. Although these techniques are orthogonal to our proposed work, and the integration of these techniques into our active learning method is beyond the scope of this paper, we will discuss the potential benefits of these techniques in Appendix F.
> > > >
> > > >
> > > > “These costs are only significant in the large scale settings (ImageNet) or on challenging tasks (segmentation), so what we call "low budget" regime on CIFAR-10 is very unrealistic anyway”
> > > >
> > > > * We would like to point out that by studying the low budget regime on CIFAR-10, we aim to design algorithms that are effective for other data sets as well in which labeling is expensive, and not just for CIFAR-10.
> > > >
> > > > &nbsp;
> > > >
> > > > ### 5. I would improve my score if the authors clearly emphasize the limitations of the experimental section
> > > >
> > > > * Thanks again for pointing out those reference papers. We will add the “Limitation and Future Work” section in Appendix F, in which the potential impacts of other factors, such as consistency regularization etc., will be discussed.
> > > >
> > > >
> > > > &nbsp;
> > > >
> > > > ### Reference:
> > > >
> > > > [1] Zhuoming Liu, Hao Ding, Huaping Zhong, Weijia Li, Jifeng Dai, and Conghui He. Influence selection for active learning. arXiv preprint arXiv:2108.09331, 2021.
> > > >
> > > > [2] Jordan T. Ash, Chicheng Zhang, Akshay Krishnamurthy, John Langford, and Alekh Agarwal. Deep batch active learning by diverse, uncertain gradient lower bounds. InInternational Conference on Learning Representations, 2020.
> > > >
> > > > [3] https://github.com/JordanAsh/badge/blob/master/query_strategies/strategy.py

---

> > > > > ### Comment · Reviewer_YgGb · 2021-11-21
> > > > > **Reply to Rebuttal**
> > > > >
> > > > > Thanks for your reply!
> > > > >
> > > > > 1. I still don't understand your response to my question about the 35% vs 44%. `In their work, querying new samples at which epoch is manually selected. Although elaborately selecting query epoch can make the ResNet achieve 45% under their hyperparameters and 44.3% under our hyperparameters, we don’t think that way is general enough because there is typically no validation set in the active learning setup, and selecting a good query epoch may need to rely on the test set.` What are you calling "query epoch"? Also why is there no validation set in the active learning setup, wouldn't it be cheaper to keep a small validation set that prevents overfitting and boosts performance by 10 percentage points?
> > > > >
> > > > > 2. `Although these techniques are orthogonal to our proposed work, and the integration of these techniques into our active learning method is beyond the scope of this paper...` I disagree that self/semi/transfer learning are outside the scope of this paper. In fact, I would even go as far as considering them essential. The proposed algorithm is accompanied by a nice theoretical inspiration, but there are massive leaps from the theory to the actual implementation (narrow networks, pseudo-labels etc..) , so the empirical section of this paper is no less important than the theoretical section. I've already mentioned 3 prior works that discuss the unexpected impacts that realistic performance boosting techniques, such as self/semi/transfer learning or advanced data augmentation techniques can have on AL algorithms. These techniques are almost always available in practice for image classification tasks and cost less time and money than labeling more data. [1] and [3] argue that AL is only useful if it can indeed provide any material boosts to performacne when combined with at least one of these techniques.
> > > > > Finally, it's worth pointing out that trying SSL or transfer learning only amounts to downloading pretrained networks and would not have presented any extra computation overhead in producing this paper.
> > > > >
> > > > >
> > > > >
> > > > > [1] Best Practices in Pool-based Active Learning for Image Classification, Submitted to The Tenth International Conference on Learning Representations 2022, https://openreview.net/forum?id=7Rnf1F7rQhR, under review
> > > > >
> > > > > [2] Mittal, Sudhanshu et al. “Parting with Illusions about Deep Active Learning.” ArXiv abs/1912.05361 (2019): n. pag.
> > > > >
> > > > > [3] Yao-Chun Chan, Mingchen Li, andSamet Oymak, On the Marginal Benefit of Active Learning: Does Self-Supervision Eat Its Cake?

---

> > > > > > ### Author Response · Authors · 2021-11-21
> > > > > > **Thanks for Reply**
> > > > > >
> > > > > > Thanks for your reply!
> > > > > >
> > > > > > 1.1 What are you calling "query epoch"?
> > > > > >
> > > > > > * The “query epoch” used in the previous reply refers to the training epoch at which the acquisition function will be used to query new samples.
> > > > > > * In the paper [1], the active method will query new samples after training 12 epochs for Faster R-CNN in the object detection task, and 200 epochs for ResNet-18 in the image classification task.
> > > > > >
> > > > > > 1.2 Why is there no validation set in the active learning setup?
> > > > > >
> > > > > > * For the small initial budget setup, splitting the small initial budget into a tiny validation set and a smaller initial training set may even hurt the model performance.
> > > > > > * For our setting, the initial budget size M=500 and for [2] M=100, then the validation set size will be 50 and 10 separately (10% as validation). The tiny validation set cannot provide enough guidance for hyperparameter selection. Therefore, we follow [2] and use the training accuracy to decide when to query new samples.
> > > > > >
> > > > > > 1.3 Wouldn't it be cheaper to keep a small validation set that prevents overfitting and boosts performance by 10 percentage points?
> > > > > >
> > > > > > * As discussed in the above comment, keeping a small validation always leads to a smaller training set which will further degrade the quality of representation. More importantly, a tiny validation set cannot provide enough guidance for hyperparameter tuning. Therefore, it is hard to guarantee the performance will be boosted by 10 percentage points.
> > > > > >
> > > > > > &nbsp;
> > > > > >
> > > > > > 2.We acknowledge the importance of the experimental section. The goal of the experimental section is to verify the utility of the proposed acquisition function and the Figure 3 and 6 are able to show the effectiveness of the proposed acquisition function by comparing it with other acquisition functions. The pre-trained model, self/semi/transfer learning, or advanced data augmentation techniques are helpful for image classification tasks and even in general helpful for a lot of computer vision tasks. However, we are focusing on the general active learning method and the proposed acquisition function is task agnostic. As the first work trying to solve the active learning problem with model dynamics, we will focus on the theoretical analysis and design of the acquisition function, and leave the discussion of the combination of the proposed method with self/semi/transfer learning techniques and even advanced image generation techniques in harder computer vision tasks in the future. Thanks again for the papers you pointed out, we have included the discussion of the potential impacts of self/semi/transfer learning techniques in Appendix F.
> > > > > >
> > > > > >
> > > > > >
> > > > > > &nbsp;
> > > > > >
> > > > > > We hope our answers address your concerns!
> > > > > >
> > > > > > &nbsp;
> > > > > > &nbsp;
> > > > > >
> > > > > > Reference:
> > > > > >
> > > > > > [1] Zhuoming Liu, Hao Ding, Huaping Zhong, Weijia Li, Jifeng Dai, and Conghui He. Influence selection for active learning. arXiv preprint arXiv:2108.09331, 2021.
> > > > > >
> > > > > > [2] Jordan T. Ash, Chicheng Zhang, Akshay Krishnamurthy, John Langford, and Alekh Agarwal. Deep batch active learning by diverse, uncertain gradient lower bounds. In International Conference on Learning Representations, 2020.

---

> > > > > > > ### Comment · Reviewer_YgGb · 2021-11-21
> > > > > > > **Response to Rebuttal**
> > > > > > >
> > > > > > > Thanks again for your reply.
> > > > > > >
> > > > > > > 1. Without validation set, the model is clearly overfitting the small amount of labeled data, since when you add validation it's possible to achieve a 25% performance boost  (35% vs 44%). Do you have experiments to backup this claim:  `The tiny validation set cannot provide enough guidance for hyperparameter selection.`? I don't think it's obvious that a small dataset won't at least reduce the effects of overfitting if you use it to identify the best validation acc during training and use that checkpoint to query.
> > > > > > >
> > > > > > > 2. Also, as a side note, in the larger budget regime, I have trained a ResNet18 with 5k randomly sampled training images and another 5k randomly selected images for validation (i.e., 10k images total.) The network can achieve 80% accuracy when trained on the 5k training images and the best ckpt is identified using the 5k validation images. In the table you provide in your response, with 5k training images, you only manage to achieve 58%.
> > > > > > > To be precise, I am referring to the first column `random`, last row `9`, of this table `CF10+ResNet, M=500, b=500, lr=0.005` in your replies. That's a 22 percentage point difference in performance (80% vs 58%) between using a good validation set and not using one at all. Surely the 5k samples that I used for validation are too many if the paper's operating in a "low budget regime", but it's not at all obvious that using a smaller 1k samples for validation wouldn't provide similar boosts, and that would totally fit in your low budget regime.
> > > > > > >
> > > > > > > At this point, I worry that most of the experiments in the paper overfit the data.

---

> > > > > > > > ### Author Response · Authors · 2021-11-22
> > > > > > > > **Thanks for Reply**
> > > > > > > >
> > > > > > > > Thanks again for your reply!
> > > > > > > >
> > > > > > > > &nbsp;
> > > > > > > >
> > > > > > > > In our experiments, as we mentioned in the previous reply and Appendix E.5, we follow [1] to query samples when training accuracy is greater than 99%. We think it is a reasonable strategy in a "low budget regime". Note, in [1], they use extensive experiments to verify that with the initial budget size M=100. Therefore, we believe that severe overfitting doesn’t happen in our experiments. Besides, the same strategy is applied to all the other active learning methods. Therefore, it is a fair comparison in our experiments.
> > > > > > > >
> > > > > > > > The 44% (the accuracy we report in the first reply) is the best performance that is achieved under our hyperparameters. The goal is to show that the learning rate schedule is not the bottleneck and address the concern “ you're trying to use the same learning rate schedule at round 0 as you are on round 9, but the ...”.  It is unfair to directly compare the results achieved with the query strategy (when to query) of [1] and the results with the query strategy of [2].
> > > > > > > >
> > > > > > > > &nbsp;
> > > > > > > >
> > > > > > > > We hope our answers address the concern!
> > > > > > > >
> > > > > > > > &nbsp;
> > > > > > > > &nbsp;
> > > > > > > >
> > > > > > > > Reference:
> > > > > > > >
> > > > > > > > [1] Jordan T. Ash, Chicheng Zhang, Akshay Krishnamurthy, John Langford, and Alekh Agarwal. Deep batch active learning by diverse, uncertain gradient lower bounds. In International Conference on Learning Representations, 2020.
> > > > > > > >
> > > > > > > > [2] Zhuoming Liu, Hao Ding, Huaping Zhong, Weijia Li, Jifeng Dai, and Conghui He. Influence selection for active learning. arXiv preprint arXiv:2108.09331, 2021.

---

### Official Review · Reviewer_p3z9 · 2021-11-02

**Correctness:** 3
**Technical Novelty And Significance:** 3
**Empirical Novelty And Significance:** 2
**Recommendation:** 6
**Confidence:** 3

**Main Review:**

This paper has a nice derivation of the dynamics of the training loss and provides an algorithm to maximize the rate of convergence.

My biggest concern is the empirical evaluation. In the training regime on CIFAR10, only 37% test accuracy is attained which is really really low. Secondly, it is strange to have different experimental settings (batch size, number of rounds) for different datasets. I think the experiments require a larger amount of data (at least up to 10k labeled points for CIFAR10 and SVHN).

Other comments:

In equation (15), is the gradient squared supposed to be in the summation? It doesn't depend on the summation so it seems like the summation is a factor of |S|.

Should the inequality in equation (23) be in the other direction? How does maximizing over a restricted (smaller) set result in a larger value?



**Summary Of The Paper:**

This paper provides a theoretically-motivated active learning algorithm based on maximizing convergence speed, which is justified by the "train faster, generalize better" intuition. The paper has limited but favorable empirical results.

**Summary Of The Review:**

The theory in this paper appears good, though I am not very familiar with NTK analyses which makes me hesitate to say more. The empirical evaluation lacks quality however.

---

> ### Author Response · Authors · 2021-11-17
> **Author Response to Reviewer p3z9**
>
> Thank you for your constructive comments!
>
> ### 1.Concern in the empirical evaluation
>
> “In the training regime on CIFAR10, only 37% test accuracy is attained which is really really low.”
>
> * For easy reference of other reviewers and ACs, we would like to clarify that the experiments pointed out by reviewer p3z9 is CNN on CIFAR10 with batch size=250, initial label set size M=500 (the left of Figure 3). The reasons for the relatively low performance are:
>   * We are using the non-retraining active learning setting, in which the model will not be trained from scratch after each query. Therefore, it is not fair to compare the results achieved in our setting and in the retraining active learning setting. Furthermore, the non-retraining setting is practical because large deep learning models cannot always afford to train from scratch in many real-world applications [1].
>   * In that experiment, we are using the vanilla CNN model. Its capacity cannot compare with other commonly used models, such as VGG and ResNet.
>   * The query batch size (b=250) in that experiment is the smallest in the whole experiments. More settings have been shown in Appendix E.2, such as a larger batch size on CNN + CIFAR10 which can achieve better performance.
>
> &nbsp;
> “It is strange to have different experimental settings (batch size, number of rounds) for different datasets.”
>
> * We evaluate the performance of dynamicAL with different experimental settings for each data set, e.g., batch size (as you mentioned) and target model. The results (Figure 3 and Appendix E.2) show our method’s effectiveness under a wide range of settings. However, due to the limited space, we only present the part of the experiment results in Figure 3, in which the results from three different settings are shown. We would like to refer the reviewer to Appendix E.2 for the remaining part of the experiment results, in which results of different batch sizes and target models on the same data set are shown.
> * To further address the concern, as suggested by the reviewer (“I think the experiments require a larger amount of data”), we increase the number of query rounds from 4 to 9 and present the results of b=500 in Figure 3 of the revision. As shown by the results in Figure 3 and Appendix E.2, our method can still systemically outperform baselines.
>
> &nbsp;
>
> ### Other comments:
>
> “In equation (15), is the gradient squared supposed to be in the summation? It doesn't depend on the summation so it seems like the summation is a factor of |S|.”
>
> * Thanks for pointing this out. We have simplified and updated Equation (15). Besides, to make sure the deposition is accessible, the derivation of Equation (15) has been provided. We would like to refer the reviewer to Appendix A.5 for more derivation details.
>
> “Should the inequality in equation (23) be in the other direction? How does maximizing over a restricted (smaller) set result in a larger value?”
>
> * The direction of inequality should be the current one, according to the Theorem 2.1 of [2]. To further demonstrate the correctness of the inequality, we provide the derivation details in Appendix B.3.
>
> &nbsp;
>
> ### Reference:
>
> [1] Ostapuk, Natalia, Jie Yang, and Philippe Cudré-Mauroux. "Activelink: deep active learning for link prediction in knowledge graphs." The World Wide Web Conference. 2019.
>
> [2] Zheng Wang and Jieping Ye. Querying discriminative and representative samples for batch mode active learning. ACM Transactions on Knowledge Discovery from Data, 9(3):1–23, 2015.

---

> > ### Comment · Reviewer_p3z9 · 2021-11-20
> > **Reply to response**
> >
> > Thank you for the response and for the additional experiments!
> >
> > I'm a bit concerned about active learning methods that only achieve gains in low accuracy regimes for whatever reason, since low performing models would likely not be practically used. I think I originally missed that you are not retraining the model after each batch. Looking at the table you provided for a different reviewer, it looks like CIFAR10 with ResNet gets nearly a 10% boost from retraining (49% -> 58%). Not only that, but the gains of dynamicAL are nearly erased when retraining (59.62% for coreset, 60.35% for dynamicAL). I'm concerned that dynamicAL has meager, if any, empirical gains over other methods in realistic settings.
> >
> > In summary, although the method introduced in this paper is derived from a nice theoretical framework, I'm concerned that the experiments may not bear out the effectiveness of this method.

---

> > > ### Author Response · Authors · 2021-11-20
> > > **Thank you for the updated feedback**
> > >
> > > Thanks for your appreciation of the theoretical novelty! Hope our reply and more experimental details (Appendix E.5) can make the empirical contribution clear.
> > >
> > > “it looks like CIFAR10 with ResNet gets nearly a 10% boost from retraining ”
> > >
> > > * Retraining can bring some benefits for the performance. However, in some real-world applications, training deep learning models from scratch is not affordable[1]. Therefore, the study of active learning in the non-retraining setting is also meaningful. The effectiveness of dynamicAL in the non-retraining setting has been shown in Section 5.2.
> > >
> > >
> > >
> > > “the gains of dynamicAL are nearly erased when retraining”
> > >
> > > * The acquisition function of dynamicAL is based on the dynamic analysis (Equation 8) which considers the change of dynamics according to the model's current parameters. Therefore, we expect the performance boosts to be more significant in the non-retraining setting. Besides, if retraining from scratch is affordable in a certain application, the performance of our method is comparable with state-of-the-art techniques, as shown by the retraining setting experiments.
> > >
> > >
> > >
> > > The non-retraining active learning setting is commonly seen in practice, and tends to be more challenging than other settings. And our experiments show the effectiveness of dynamicAL in the non-retraining setting with different target models and datasets. However, if retraining from scratch is affordable in a certain application, the performance of our method is comparable with state-of-the-art techniques, such as Coreset [2]. We include the experiments of retraining settings in Appendix E.5 and provide a discussion about the empirical results.
> > >
> > >
> > >
> > > **Reference:**
> > >
> > > [1] Ostapuk, Natalia, Jie Yang, and Philippe Cudré-Mauroux. "Activelink: deep active learning for link prediction in knowledge graphs." The World Wide Web Conference. 2019.
> > >
> > > [2] Ozan Sener and Silvio Savarese. Active learning for convolutional neural networks: A core-set approach. In International Conference on Learning Representations, 2018.

---

### Official Review · Reviewer_8dDv · 2021-11-02

**Correctness:** 4
**Technical Novelty And Significance:** 3
**Empirical Novelty And Significance:** 3
**Recommendation:** 6
**Confidence:** 3

**Main Review:**


The work is very timely, since it leverages on recent results to provide a better objective for deep active learning, namely the time independence of the NTK and the train-faster--generalize-better paradigm. From there, everything descends smoothly, providing a clean story and a method that seems to work well. Even though the theoretical analysis is only justified in the ultra-wide limit, the empirical evidence provided (on satisfactory models and datasets) suggests that the method is still good for typical use cases.

As the authors truthfully recognize, with active learning the data is not iid, and this needs to be addressed. In particular, it is not clear that the dominant term of the population risk is given by B (which then goes as 1/Alignment). To do so, they upper bound the difference of the population risks using the two distributions (from population and from active learning) by the Maximum Mean Discrepancy (MMD). Then, they show empirically that MMD<B for several values of b, on the first query round:

- They state that  the MMD is always much smaller than B, but the blue and yellow bars in figure 1 look similar. It took me a long while to realize that the bars have different axes. There is no legend nor visual aid that helps the reader. This must be changed.

- They only show this for round R=1. Could one expect that the two quantities approach each other for larger R? My intuition would say that the MMD increases with R (though I don't know by how much).

- Practitioners would be often interested in quite large R, but results do not seem to go in this sense.

- From figure 1 we see that bigger b implies a smaller B/MMD. This can be expected, since with a bigger b the two distributions are more different. Would it be more interesting to compare at constant budget size, as done in table 1?

- If we then look at table 1, we see that the best performing schemes are those with many smaller query rounds. Is there some intuitive reasoning that could lead to this expectation? What is the recommendation that the authors would give to a practitioner who wants to implement their method?

Did the authors observe a deterioration in their active learning scheme as models become narrower? Should we expect this? Does this mean that dynamicAL should not be used for "deep and narrow" models?

Why only 4 query rounds for the CIFAR and Caltech datasets? It doesn't seem that the performance has converged as for SVHN (which got 9 rounds), and the performance is not close to those that are obtained in practice.

Figure 3 might be more readable if different symbols are used for each method. I had a hard time identifying each curve.

There's a couple of typos and expressions that need some English polishing (nueral, convergence faster, ...).


**Summary Of The Paper:**

The authors address the problem of active learning in the context of deep learning. Instead of querying new examples based on the decision boundary (which in nonlinear models can be tricky or even ill-defined), as it is usually done for linear models, they rely on the train-faster--generalize-better paradigm. Thus, they propose to optimize the "training dynamics", which is the time derivative of the loss function in the ultra-wide limit. By using pseudo labeling and subset approximation, their method allows for fast selection of the examples that should be added to the dataset.
They justify their active learning strategy through an analysis in the limit of very wide models, through the neural tangent kernel, which is particularly convenient since the NTK does not depend on time in the ultra-wide limit.
They study a quantity that they call "alignment" (which measures the correlation between input and output in the NTK space) and show that a higher alignment is related to better generalization (a better bound) and faster training (--> larger training dynamics). They then show that this is also true in the active-learning setting, where the data is no more iid, with the help of some empirical evidence. Finally, they test their method and compare it with other active learning strategies, which they systematically outperform.


**Summary Of The Review:**

The paper looks good. Some aspects of the presentation can be improved. Some of the results rely on approximations or numerical evidence which might not be universal, but I find the method quite interesting, and it paves the road to new possibilities in deep active learning.

---

> ### Author Response · Authors · 2021-11-17
> **Author Response to Reviewer 8dDv**
>
> Thank you for your constructive comments!
> &nbsp;
>
> ### 1. The comparison between MMD and B
> &nbsp;
> “It took me a long while to realize that the bars have different axes. There is no legend nor visual aid that helps the reader. This must be changed.”
> &nbsp;
>
> * Thank you for the suggestion. Figure 1 has been updated with legend and colorized axis.
> &nbsp;
>
> “Could one expect that the two quantities approach each other for larger R? My intuition would say that the MMD increases with R (though I don't know by how much).”
> &nbsp;
>
> * The MMD term may not always monotonically increase with the increase of query rounds R. It depends on which data is sampled. Specifically, the MMD term induced after the second round might be smaller than the term after the first round, because the sampled data in the second round can make the empirical data distribution closer to $p(x)$. **Therefore, we should not always expect the two quantities to be closer to each other for larger R.** To better answer the question, the empirical measurement for a larger R is necessary and the result for more query rounds is shown in Figure 7, Appendix E.4. The MMD is much smaller than B even for a large R.
> &nbsp;
>
> “Would it be more interesting to compare (B/MMD) at constant budget size, as done in table 1?”
> &nbsp;
>
> * Thanks for the suggestion. We measure B/MMD at the constant budget size and the result is in Table 10, Appendix E.4. At R=0, a larger batch size will lead to larger MMD/B, because the sampling bias of the query policy is amplified by using a large batch size. However, it may not always be that case for R > 1, because the data (biased) sampled in another round may make the empirical distribution of the whole sampled data closer to the data distribution.
> &nbsp;
>
> “If we then look at table 1, we see that the best performing schemes are those with many smaller query rounds. Is there some intuitive reasoning that could lead to this expectation?”
> &nbsp;
>
> * More query rounds allow dynamicAL to have more chances to update its estimation for unlabeled data samples (with better Pseudo Labeling and Subset Approximation) before running out of the total budget. Therefore, for a constant budget size, many smaller query rounds in general perform better.
> &nbsp;
>
> “What is the recommendation that the authors would give to a practitioner who wants to implement their method?”
> &nbsp;
>
> * Considering the time limitation in the real-world scenario, a practitioner may first try to use relatively more query rounds. Take the CF10 as an example. If the total labeling budget is 1000, then b=100, R=10 is a good starting point. Relatively small batch size can make dynamicAL have more chance to update its estimation and query more informative samples in the process of active learning. Note that it doesn’t mean setting b=1 will be a good option. As shown in Figure 4, dynamicAL can have a good subset approximation for large batch sizes. Therefore, using an extremely small batch size will introduce extra unnecessary computation.
>
> &nbsp;
> &nbsp;
>
> ### 2.dynamicAL for "deep and narrow" models
> &nbsp;
> “Did the authors observe a deterioration in their active learning scheme as models become narrower? Should we expect this? Does this mean that dynamicAL should not be used for "deep and narrow" models?”
>
> * In our experiments, we use the vanilla CNN (narrower than VGG and ResNet) as the target model, in which our AL method is still effective. In other words, we didn’t observe a deterioration as the target model became narrower in our experiments.
>   Recent works have shown the effectiveness of NTK theoretical analysis for commonly used model architectures. For example,
>     - Park et al. [1] used the NTK to predict the generalization performance of architectures in the application of Neural Architecture Search (NAS).
>     - Chen et al. [2] used the condition number of NTK to predict a model’s trainability.
>     - Chen et al. [3] also used the NTK to evaluate the trainability of several ImageNet models, such as ResNet.
>     - Deshpande et al. [4] used the NTK for model selection in the fine-tuning of pre-trained models on a target task.
>
> &nbsp;&nbsp;&nbsp;&nbsp;&nbsp;&nbsp;&nbsp; Based on these results, we do not expect deterioration in performance for commonly used models.
>
> * We acknowledge that the theoretical study for the “deep and narrow” network is meaningful, such as the study about under what condition the original analysis will be ineffective. However, we don't believe the extremely deep and narrow networks are practical. On the contrary, as demonstrated in [1][2][3][4] and further verified by our experiments,  the analysis derived based on NTK is valid for the commonly used practical models.

---

> > ### Author Response · Authors · 2021-11-17
> > **Author Response to Reviewer 8dDv (Cont.)**
> >
> > ### 3. Experiment in CIFAR and Caltech datasets
> >
> > “Why only 4 query rounds for the CIFAR and Caltech datasets? ”
> >
> > * We actually ran 9 rounds for CIFAR with ResNet (Appendix E.3). For CIFAR + CNN and Caltech + ResNet, we think 4 rounds are enough to show the effectiveness of our methods. To further address the concern, we ran the experiments on CIFAR + CNN and Caltech + ResNet with 9 rounds and updated the corresponding figures and tables. As shown in Figure 3 of the revision, dynamicAL can systematically outperform those baseline methods in different settings.
> >
> > &nbsp;
> > &nbsp;
> >
> > ### 4. Suggestions: Figure 3 with different symbols; typos and expressions
> >
> > * Thanks for pointing this out and helping us improve the manuscript. We have updated Figure 3 with different symbols for each method and fixed typos in the revision.
> >
> > &nbsp;
> > &nbsp;
> >
> > ### Reference:
> >
> > [1] Daniel S Park, Jaehoon Lee, Daiyi Peng, Yuan Cao, and Jascha Sohl-Dickstein. Towards nngp guided neural architecture search. arXiv preprint arXiv:2011.06006, 2020.
> >
> > [2] Wuyang Chen, Xinyu Gong, and Zhangyang Wang. Neural architecture search on imagenet in four gpu hours: A theoretically inspired perspective. In International Conference on Learning Representations, 2021a.
> >
> > [3] Chen, Xiangning, Cho-Jui Hsieh, and Boqing Gong. "When Vision Transformers Outperform ResNets without Pretraining or Strong Data Augmentations." arXiv preprint arXiv:2106.01548 (2021).
> >
> > [4] Deshpande, A., Achille, A., Ravichandran, A., Li, H., Zancato, L., Fowlkes, C., Bhotika, R., Soatto, S. and Perona, P., 2021. A linearized framework and a new benchmark for model selection for fine-tuning. arXiv preprint arXiv:2102.00084.

---

> > > ### Comment · Reviewer_8dDv · 2021-11-17
> > > **Thanks for your reply**
> > >
> > > I thank the authors for addressing my comments.
> > >
> > > Just as a note, I like the new material in appendix E.4, but I do not see it referenced in the text.

---

> > > > ### Author Response · Authors · 2021-11-18
> > > > **Thanks Again**
> > > >
> > > > We are glad that our clarification and the additional details are helpful, and we have updated the revision and included the reference for Appendix E.4. Thank you very much for your time and helping us to improve the manuscript!

---

### Author Response · Authors · 2021-11-25
**To All Reviewers: Updated Manuscript and Appendix**

Dear Reviewers,
&nbsp;

We thank the reviewers for their comments and suggestions that helped us to improve the manuscript. For easy reference of reviewers and ACs, all changes have been highlighted in blue. Some of the main changes include:

1. Provided the derivations of Equation (13) and (15) in Appendix A.4 and A.5 respectively.
2. Provided the derivations for Equation (23) in Appendix B.3.
3. Changed the legend and colorized the axis of Figure 1.
4. Provided additional experiments (Figure 7 and Table 10) and a discussion about the relationship between MMD and B in multiple rounds in Appendix E.4.
5. Added a further discussion about the experimental result in Section 5.2.
6. Updated Figure 3 and Figure 6 with larger total budget size and total query rounds.
7. Added a discussion for the performance in the retraining setting in Appendix E.5 with experiments (Table 11 and Table 12).
8. Added the “Limitation and Future Work” section in Appendix F, in which the potential impacts of other techniques and factors, such as consistency regularization, data augmentation, large budget size, have been discussed.
9. Changed symbols used for each method in Figure 3 and Figure 6.
10. Corrected several typos.

We appreciate all reviewers for the hard work and helpful comments. We would like to address all reviewers’ concerns in the corresponding responses.
&nbsp;

Sincerely,

Authors of Paper 2516

---

### Decision · Program_Chairs · 2022-01-20

**Decision:**

Reject

**Comment:**

This paper proposes a novel strategy for deep active learning based on the training dynamics of the underlying deep model, defined as the derivative of the loss of the ultra-wide NTK. All reviewers enjoy the clean story and motivation of the proposed acquisition/objective function and appreciate the authors’ effort in providing theoretical justification and analysis.

One note is that -- as Reviewer 8dDv highlighted -- part of the analysis pertaining to the incompatibility between the generalization bound of NTK and the non-iid nature of active learning rely on numerical evidence: the MMD under the covariate shift setting (i.e. assuming that the conditional distributions P(Y |X) remains consistent) is shown empirically to be smaller than the dominant term of the generalization bound. This serves as a reasonable empirical motivation/ justification of the dynamicAL heuristic under the AL setting, but I would suggest the authors be more precise in the abstract / intro (e.g. abstract) that this is an empirical result.

While the theoretical results are interesting, not all reviewers are convinced that the experimental results are sufficiently compelling. In particular, Reviewer YgGb points out that the significant performance boost reported in the main paper was mainly due to the non-retraining (i.e. not retraining the model (from scratch)) constraints imposed by the problem setup. Reviewer p3z9 shares the same concern that such a setting would be far from realistic at least for the data sets/labels considered in the experiments. The authors refer to Ostapuk et al, 2018 as a justification of the non-retraining setting; yet they assumed a high budget, e.g., up to 50% of all labels of datasets.

In summary, this is a theoretically well-motived work, but the empirical components need to be further clarified and supported with more realistic experiments to merit acceptance for the proposed solution.